# Heterogeneity in responses to ribosome-targeting antibiotics mediated by bacterial RNA repair

Hollie J. Hindley [1], Zechuan Gong[2], Shafagh Moradian [2], Maria Grazia Giuliano[2,3], Andrei Sapelkin [4], Ioly Kotta-Loizou [5,6], Martin Buck [5], Christoph Engl [2] ✉ & Andrea Y. Weiße [1,7] ✉

RNA repair is critical for cellular function. The Rtc system maintains RNA integrity within the translational machinery of bacteria. In *E. coli*, Rtc expression enables cells to rescue growth and survive treatment by conferring transient resistance to ribosome-targeting antibiotics, yet the mechanisms underpinning this resistance remain obscure. Here, we present a computational model of Rtc-regulated repair of translational RNAs. Integrating model predictions with experimental validations, we uncover notable cell-to-cell heterogeneity in *rtc* expression that impacts on translational capacity, indicating that *rtc* may induce a form of heteroresistance. We moreover identify Rtc targets that may reduce the translational capacity of cells and so potentiate antibiotic effects. Our findings elucidate a complex response underpinning resistance conferred by Rtc, offering alternate routes for addressing resistance in *E. coli* and other relevant pathogens.

RNA is an essential macromolecule in all living cells. Aside from messenger RNA (mRNA), transfer RNA (tRNA) and ribosomal RNA (rRNA), all of which are elemental to protein expression, additional RNA species play significant roles within cells, such as ribozymes, microRNAs and small nuclear RNAs contributing to catalysis, gene regulation and intron splicing, respectively[1]. RNA is susceptible to damage by processes such as alkylation, radiation and oxidation, in addition to RNA-specific damaging agents such as ribonucleases and ribotoxins[2]. As most abundant types of RNA are long-lived[3], damage to them has potentially enduring effects. With repair likely to require less energy than de novo synthesis, RNA repair can provide an evolutionary advantage to cells[4]. To maintain vital functions, cells have therefore developed specific RNA repair mechanisms, such as AlkB, which repairs alkylative RNA damage in both humans and *E. coli* [5,6].

Here we investigate two RNA repair proteins, the RNA cyclase RtcA and the RNA ligase RtcB, which are conserved across all domains of life[7]. In metazoans and archaea, RtcB plays a well-defined role in tRNA ligation after intron excision and in the unfolded protein response[8,9]. In bacteria, however, the exact roles of Rtc are less known. Previous work on *E. coli* showed that Rtc plays roles in chemotaxis and motility[10]. In addition, Rtc has been shown to be key for maintenance of the translational apparatus of *E. coli*, where expression of Rtc is induced by oxidative damage or impairments to the translation apparatus, specifically, ribosome-targeting antibiotics and other translation-inhibiting agents such as colicin D[10]. The ligase RtcB has been implicated with tRNA and rRNA repair: increased degradation of tRNA in the absence of functional RtcB suggests that RtcB contributes to the repair of damaged tRNAs[11–13], while other evidence suggests that RtcB also improves ribosome health through the re-ligation of

[1]Centre for Engineering Biology, School of Biological Sciences, University of Edinburgh, Edinburgh, UK. [2]Department of Biochemistry, Centre for Molecular Cell Biology, School of Biological and Behavioural Sciences, Queen Mary University of London, London, UK. [3]Health Science Interdisciplinary Center, Sant'anna School of Advanced Studies, Pisa, Italy. [4]Department of Physics and Astronomy, School of Physical and Chemical Sciences, Queen Mary University of London, London, UK. [5]Department of Life Sciences, Faculty of Natural Sciences, Imperial College London, London, UK. [6]Department of Clinical, Pharmaceutical and Biological Science, School of Health, Medicine and Life Sciences, University of Hertfordshire, Hatfield, UK. [7]School of Informatics, University of Edinburgh, Edinburgh, UK. ✉e-mail: c.engl@qmul.ac.uk; andrea.weisse@ed.ac.uk

fragmented rRNA with ribosomal damage accumulating in the absence of RtcB[14,15]. Another target of RtcB, the transfer messenger RNA *ssrA*, is involved in rescuing stalled ribosomes upon translation defects, further highlighting the importance of Rtc in maintaining translational activity[11,16].

While *rtcA* and *rtcB* are widely conserved, *E. coli* and some closely related species, including *Salmonella*, *Klebsiella* and *Shigella* species, share a distinct regulation[17], where the *rtcBA* genes are co-expressed from the same promoter under tight control of the regulator protein, RtcR. Transcribed divergently to *rtcBA*, RtcR completes the *rtc* locus[7] (Fig. 1A). RtcR is a bacterial enhancer binding protein (bEBP), activation of which precedes *rtcBA* transcription following exposure to inducing conditions[18,19]. Upon expression, RtcBA can repair damaged RNA in a concerted healing and sealing procedure[20], where RtcA cyclizes (heals) damaged RNA ends, and RtcB subsequently ligates (seals) cyclic phosphate termini with 5′-OH termini to repair the RNA perfectly[14] (Figure 1B).

The precise damage inflicted by Rtc-inducing antibiotics has not been fully characterised and may affect ribosomal or transfer RNAs[21–23]. Repair of the damaged RNA then enables expressing cells to restore translational activity and rescue growth amid antibiotic assault (Fig. 1C). The system thus affords cells with an intrinsic antibiotic resistance present only in cells expressing Rtc as a regulated stress response upon RNA damage[24]. The nature of Rtc-induced resistance is transient, as cells acquire resistance as and when Rtc is expressed. Therefore, cells are able to adapt to survive until the antibiotic challenge has passed or a permanent specific resistance mechanism is acquired[25]. Given the widespread use of ribosome-targeting antibiotics, which across England accounted for over 40% of clinical use in 2022[26], Rtc-induced antibiotic resistance could be contributing to the growing antimicrobial resistance problem.

Despite the importance of RNA repair across all domains of life, there has been little research on RNA repair in comparison to the more widely studied DNA repair systems. Given the uncertainty surrounding the role of Rtc within resistance, we developed a mathematical model of the *E. coli rtcBA* system to investigate the mechanistic action of RNA repair proteins in maintaining the translation apparatus and leading to induced antibiotic resistance. The model provides a computational framework to study Rtc-regulated RNA repair and gain a systemic understanding of its physiological implications.

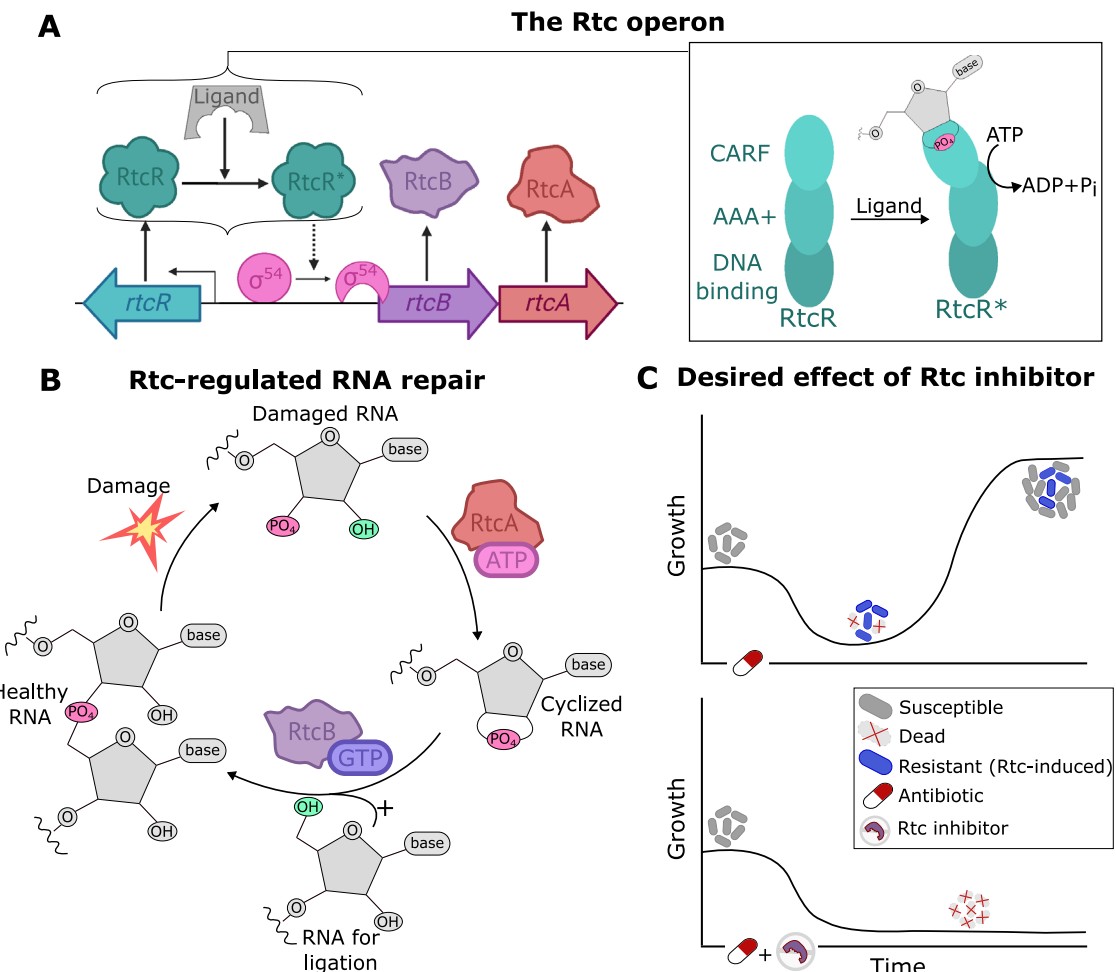

**Fig. 1 | rtcBA expression and Rtc-induced RNA repair in *E. coli*. A** The *rtc* locus contains three Rtc genes: *rtcBA* are co-expressed from a $\sigma^{54}$-controlled promoter regulated by RtcR. RtcR contains three domains (right side of panel **A**). The CARF domain imposes negative regulation on the AAA$^+$ domain; binding of an RNA with a 2′,3′-phosphate end to the CARF domain causes activation of the intrinsic ATPase activity of the AAA$^+$ domain to produce active RtcR which initiates transcription of *rtcBA*. **B** Healthy RNA, with phosphodiester bonds linking adjacent RNAs, can be damaged to produce RNA with a 3′-phosphate end (damaged RNA). RtcA requires ATP to convert RNA with 3′-phosphate termini to 2′3′-cyclic phosphate termini (cyclized RNA). RtcB requires GTP to ligate 2′3′-cyclic phosphate with RNA 5′-OH termini (RNA for ligation) to complete the cycle and repair the RNA with a perfect phosphodiester bond (healthy RNA). **C** Treatment with a ribosome-targeting antibiotic will initially cause a reduction in growth but will also cause the onset of Rtc-induced resistance. Resistant cells stay alive despite the antibiotic and resume growth rendering the antibiotic ineffective (upper panel). If infections could be treated with a ribosome-targeting antibiotic in combination with an Rtc-inhibitor, Rtc-induced resistance may be avoided (lower panel).

## A Rtc-regulated RNA repair

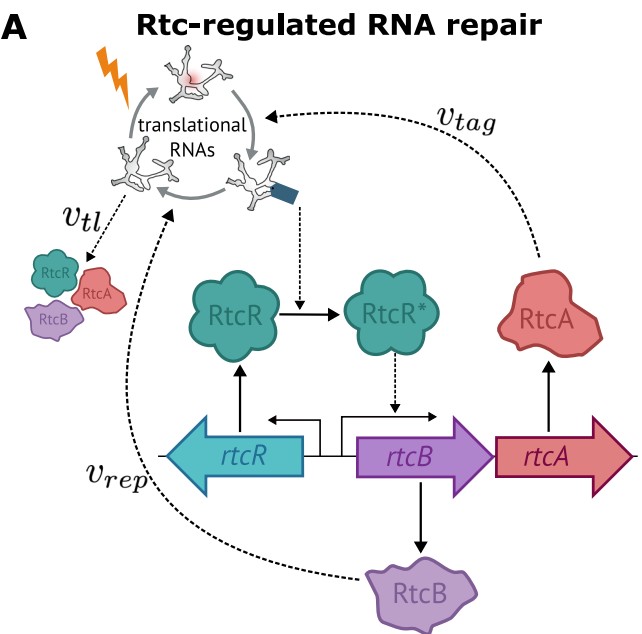

## B        Model rates and definitions

| Rate | Description | Definition |
|---|---|---|
| $v_{\text{tag}}$ | Tagging rate by RtcA | $p_A \cdot \dfrac{r_d \cdot k_{\text{tag}}}{r_d + K_a}$ |
| $v_{\text{rep}}$ | Repair rate by RtcB | $p_B \cdot \dfrac{r_t \cdot k_{\text{rep}}}{r_t + K_b}$ |
| $v_{\text{tx}}^{(R)}$ | Transcription rate of *rtcR* | $\dfrac{\omega_r \cdot \text{ATP}}{\theta_{\text{tx}} + \text{ATP}}$ |
| $v_{\text{tx}}^{(BA)}$ | Transcription rate of *rtcBA* | $\sigma_o \cdot \dfrac{\omega_{\text{ba}} \cdot \text{ATP}}{\theta_{\text{tx}} + \text{ATP}}$ |

**Fig. 2 | Model of Rtc-regulated RNA repair. A** Model schematic. The model describes the dynamic interplay of the Rtc operon with a pool of damaged, healthy and tagged ribosomes. Tagged ribosomes, which contain rRNA with 2',3'-cyclic phosphate ends, act as the ligand for RtcR to activate *rtcBA* transcription. RtcA converts damaged ribosomes (containing RNA with 3'-phosphate ends) to tagged ones, and RtcB converts tagged ribosomes to healthy ones. Healthy and tagged ribosomes feed back to Rtc expression via translation and RtcR-activation, respectively. **B** Dependent rates and their definitions. Parameter descriptions and values can be found in the SI.

We perform a comprehensive analysis of the Rtc model, which reveals a previously unknown trait of the Rtc system, namely that the regulatory mechanisms governing Rtc expression promote bistability. Bistability enables cells to adopt distinct expression phenotypes that can coexist across a range of conditions. Bistable responses have been linked with other phenotypes of antibiotic resistance, where isogenic cells can either be susceptible or resistant within the same environment[27]. The model predicts that cells expressing Rtc are able to counteract the damage and recover translational activity, whereas translation collapses in non-expressing cells. We provide experimental evidence which validates model predictions on cell-to-cell heterogeneity in *rtc* expression, and moreover, that expressing cells display higher translational activity upon exposure to a ribosome-targeting antibiotic. The findings suggest that Rtc-induced resistance to ribosome-targeting antibiotics may constitute a form of heteroresistance[28].

Using the Rtc repair model, we computationally screen for molecular targets amenable to suppressing Rtc expression and locking cells in the susceptible state (Fig. 1C). Our computational analysis suggests that inhibition of the ligase RtcB as well as the transcriptional

regulator RtcR result in effective suppression of RNA repair and translational recovery. Targeting the cyclase RtcA, in turn, only results in a reduction of Rtc expression, where cells can still recover translational activity. Motivated by these predictions, we then quantify ribosome concentration, as a readout of translational capacity, across single *E. coli* cells that were challenged with the ribosome-targeting antibiotic tetracycline. We find that cells lacking *rtcB* or *rtcR* indeed display a more severely affected translational capacity compared to those lacking *rtcA*. The data thus provide initial validation that RtcR or RtcB, rather than RtcA, present promising targets for inhibition aimed at reducing levels of heteroresistance.

Our work revealed a previously unknown form of resistance to commonly used antibiotics and thus inspires research to optimise usage of these drugs. The results highlight the potential of the computational toolkit we developed to generate testable hypotheses. We further identified specific molecular targets that may be inhibited in combination with ribosome-targeting antibiotics to increase the overall effectiveness of treatment. Considering the broad use of Rtc-inducing antibiotics and the high level of conservation of *rtcBA* across several clinically relevant bacterial pathogens, our results pinpoint a potential avenue to address the incidence of antibiotic resistance.

## Results

### A dynamic model of Rtc-regulated RNA repair

We developed a computational model of Rtc-regulated RNA repair in *E. coli* that describes the regulated expression of Rtc proteins and their dynamic action to heal and seal damaged RNA within the translation machinery (Fig. 2A). We model the expression of the three *rtc* genes by considering their transcription to mRNA and subsequent translation to proteins. Throughout, we assume that Rtc proteins act to repair damaged rRNA, but we also show that the same conclusions hold when Rtc maintains damaged tRNA (see SI). To this end, we model three distinct species of ribosomes: healthy ribosomes that can obtain breakage within their rRNA and become damaged ribosomes; upon interaction with RtcA, damaged ribosomes convert to what we call 'tagged' ribosomes, which are still considered to be damaged but have been pre-processed for final repair. These tagged ribosomes contain cyclized rRNA ends and serve as a substrate to RtcB for full repair to recover healthy ribosomes.

Overall, the model considers nine molecular species, the three mRNAs, $m_x$, where $x \in \{R, B, A\}$ stands for the different *rtc* genes, their corresponding proteins, $p_x$, as well as three species of translational RNA $r_h$, $r_d$ and $r_t$, denoting healthy, damaged and tagged RNA, respectively. Translational RNA, which we sometimes refer to by RNA, may represent rRNA or tRNA. For brevity, main figures depict results of the rRNA model and equivalent results for the tRNA model are given in the SI. We model Rtc expression and the dynamic interaction of the Rtc repair proteins with the translational machinery via a set of ordinary differential equations. Below we detail the mechanistic assumptions we use to derive the rates of change for each of the variables, starting with the regulatory control of Rtc expression and followed by the repair steps underpinning the maintenance of translational RNA.

**Rtc-regulated RNA repair.** The biochemical mechanisms of RtcA and RtcB are well understood; RtcA performs RNA-end healing while RtcB is responsible for sealing RNAs. Despite this comprehensive understanding, the link between these mechanistic actions and their physiological significance in bacteria remains unclear. RtcA converts 3'-phosphate RNA ends to 2',3'-cyclic phosphate termini (Fig. 1B). The process of cyclisation is carried out in a three-step reaction mechanism which relies upon the presence of ATP and $Mn^{2+}$ ions[17,29]. RtcA can also cyclise 2'-phosphate ends but at a much slower rate than 3' ends[17]. We primarily model RtcA action on 3'-phosphate ends, but also consider the action on 2'-phosphate ends by using the lower catalytic rate of cyclisation in the model.

RtcB ligates 2',3'-cyclic phosphate termini with 5'-OH termini to produce a phosphodiester bond and repair the RNA perfectly (Fig. 1B), in a process dependent on GTP and $Mn^{2+}$ [30–32]. It has also been shown that RtcB can directly repair 3'-phosphate ends[31], however, at a much slower rate than cyclisation by RtcA[33] (cf. Table S1). When the two are co-expressed, as in the antibiotic response we consider, RtcA thus likely removes reactive 3'-phosphate ends before RtcB can, and so in the model we only consider the action of RtcB on 2',3'-cyclic phosphate termini.

Alternatively, damage may directly yield 2',3'-cyclic phosphate RNA ends[34], in which case RtcA would not be required for repair. This scenario gives rise to much the same dynamics as the concerted repair of 3'-phosphate ends by RtcA and RtcB, because the cyclization by RtcA is extremely fast (see §S1.4.3 and Fig. S1). In addition, it has been shown that RtcA is required for RtcR activation[11], and since *rtcBA* are co-expressed, here we assume that RtcA is necessary to produce the cyclic ligand for activation and substrate for RtcB.

The *E. coli rtcBA* system is induced by agents that challenge the translation apparatus, where both tRNA and rRNA may be targeted for damage[10]. Whilst ribosome-targeting antibiotics may not directly damage translational RNA, expression of stress-induced genes, such as *mazF*[15], in response to cellular stress can cause rRNA damage. Similarly, oxidative stress can indirectly damage both tRNA and rRNA[21,35,36]. We can therefore model damage and repair on both rRNA within a ribosome and on tRNA. In the case of rRNA repair, RtcB has been shown to only interact with rRNA that is structurally intact within a ribosome[14,15], suggesting that RtcB will not repair free rRNA that has been released from collapsed ribosomes. We thus model the repair of rRNA within structurally intact ribosomes.

Figure 2A shows how we model the presence of three distinct species of translational RNA, which are controlled by the action of RtcA, RtcB and damage. We assume that only healthy translational RNA is available for translation. Upon damage, healthy RNA becomes non-functional, damaged RNA which is converted to tagged RNA via RtcA. Tagged RNA is converted back to healthy RNA via the action of RtcB.

The dynamics of healthy RNA ($r_h$), damaged RNA ($r_d$) and tagged RNA ($r_t$) are described by the following differential equations:

$$\dot{r}_h = k_{\text{in}} + v_{\text{rep}} - r_h \cdot (\lambda + k_{\text{dam}}), \quad (1)$$

$$\dot{r}_d = r_h \cdot k_{\text{dam}} - v_{\text{tag}} - r_d \cdot (\lambda + d_r), \quad (2)$$

$$\dot{r}_t = v_{\text{tag}} - v_{\text{rep}} - \lambda \cdot r_t, \quad (3)$$

where $\dot{y}$ denotes the rate of change in species $y$. Rates that further depend on molecular species or parameters we denote by $v^{(y)}_{\text{process}}$, where subscripts indicate the process, and whenever clarification is required, superscripts indicate the affected molecular variable $y$ (see Fig. 2B). We assume that there is a constant influx of healthy rRNA or tRNA at rate $k_{\text{in}}$, and that ribosome-targeting antibiotics damage healthy RNA at a constant rate $k_{\text{dam}}$. Damaged RNA contains 3'-phosphate RNA and is converted to tagged RNA through the action of RtcA with rate $v_{\text{tag}}$, or degraded with rate $d_r$. Tagged RNA contains 2',3'-cyclic termini, which is both the product of the RtcA healing reaction and the RtcR-activating ligand. Tagged RNA is converted back to healthy RNA by the ligating action of RtcB at rate $v_{\text{rep}}$. All species are diluted by growth with rate $\lambda$.

Equations (1)-(3) describe a dynamic pool of rRNA or tRNA, respectively, where different RNA states are governed by RtcA and RtcB as well as the addition of damage through antibiotic stress. Next we explain the assumptions underlying the dependence of Rtc expression upon ribosomal levels to give the full model of Rtc-regulated RNA repair (Fig. 2A).

**The rtc operon.** We consider transcription and translation of the three *rtc* genes. We assume that mRNA levels, $m_x$ with $x \in \{A, B, R\}$ for *rtcA*, *rtcB* and *rtcR*, respectively, are governed by production via transcription ($v^{(x)}_{\text{tx}}$) and decay via dilution ($\lambda$) as well as mRNA degradation ($d_m$), giving rise to the differential equations

$$\dot{m}_x = v^{(x)}_{\text{tx}} - m_x \cdot (\lambda + d_m). \quad (4)$$

Bacterial enhancer binding proteins, including RtcR, are often expressed from promoters regulated by $\sigma^{70}$, which are a family of $\sigma$-factors associated with regulating housekeeping genes[37]. We therefore assume constitutive *rtcR* expression[38], where the transcription rate $v^{(R)}_{\text{tx}}$ only depends on physiological parameters of the cell (see Figs. 2B and SI). As *rtcA* and *rtcB* share a joint promoter, we assume that $v^{(A)}_{\text{tx}} = v^{(B)}_{\text{tx}}$ at all times.

Transcription of *rtcBA* is regulated by $\sigma^{54}$[47]. Unlike $\sigma^{70}$-controlled promoters, which can initiate transcription spontaneously, $\sigma^{54}$-regulation requires ATP to remodel the holoenzyme (RNAP:$\sigma^{54}$) from the closed complex to the open complex so DNA can enter the RNAP active site[39]. RtcR provides ATP for formation of the open complex, but firstly requires its own activation[11]. Figure 1A shows the Rtc operon, including control of *rtcBA* expression through activation of RtcR, followed by the remodelling of the $\sigma$-factor containing holoenzyme.

Similar to the structure of other bacterial enhancer binding proteins, RtcR has three domains: a regulatory CARF (CRISPR-associated Rossman fold) domain, a DNA binding domain and an $AAA^+$ domain[40,41] (Fig. 1A). The CARF domain of RtcR imposes negative regulation on the $AAA^+$ domain, evidenced by over-expression of an N-terminal deletion strain where expression of *rtcBA* was constitutively active[7]. To alleviate RtcR auto-inhibition, an unknown ligand binds the CARF domain to activate the $AAA^+$ domain. This ligand could be a linear RNA molecule, a tRNA fragment or an RNA with 2',3'-cyclic termini, as CARF domains are known to bind cyclic termini[11,19,42]. Therefore, in the model we assume that tagged RNA $r_t$, i.e. the product of the RtcA reaction which contains RNA with 2',3'-cyclic termini, act as the ligand for RtcR activation (Fig. 1A).

Following ligand binding, RtcR oligomerises to form fully active RtcR with functional ATPase activity to remodel the holoenzyme for initiation of transcription of *rtcBA*[11]. Consequently, in the model we assume that RtcR acts as a hexamer, as is typical for bEBPs[43], therefore requiring the cooperative binding of up to six ligands for full activation[19,44]. We model this cooperative binding with a Monod-Wyman-Changeux (MWC) model which describes the cooperative activation of proteins made up of identical subunits[45] (see SI). By adjusting the MWC parameters, we include spontaneous activation of RtcR, and subsequent leaky transcription of *rtcBA* (see Supplementary Table S1). Therefore, when there is no damage, there can still be a baseline level of *rtcBA* expression, which accounts for the recent finding that RtcR activation requires the binding of a ligand in addition to the presence of both RtcA and RtcB[11]. The total rate of *rtcBA* transcription, $v^{(\text{BA})}_{\text{tx}}$, thus accounts for the rate of transcriptional initiation via RtcR,

$$\sigma_o = \frac{p^*_R \cdot v_{\text{oc}}}{k_{\text{diss}}}, \quad (5)$$

where $v_{\text{oc}} = \frac{V_{\text{max}} \cdot \text{ATP}}{K_m + \text{ATP}}$, as well as physiological parameters that can impact the rate of transcriptional elongation (see Table in Fig. 2B and derivation in the SI).

Finally, we model each Rtc protein, $p_x$ for $x \in \{A, B, R\}$, independently via

$$\dot{p}_x = v^{(x)}_{\text{tl}} - \lambda \cdot p_x. \quad (6)$$

Here, we assume that protein decay is dominated by dilution and active degradation is negligible. The key difference between translational RNA representing rRNA vs tRNA is how their healthy forms, $r_h$, differentially impact translation. Ribosomes scale translation linearly by

$$v_{tl,\,rRNA} = (k_c \cdot r_h \cdot m_x) \cdot \frac{1}{n_x} \cdot \frac{\gamma_{max} \cdot ATP}{\theta_{tl} + ATP}, \qquad (7)$$

where the term in parentheses accounts for binding with mRNA and the remainder for translation elongation, which is scaled inversely with the length, $n_x$, of protein $p_x$. When $r_h$ represents healthy tRNA, ribosome levels are considered constant, denoted by $R$, and $r_h$ impacts translation elongation in a saturable manner by

$$v_{tl,\,tRNA} = (k_c \cdot R \cdot m_x) \cdot \frac{1}{n_x} \cdot \frac{\gamma_{max} \cdot ATP}{\theta_{tl} + ATP} \cdot \frac{r_h}{\theta_t + r_h}. \qquad (8)$$

Equations (1)-(8) comprise the model of Rtc-regulated RNA repair. For brevity, we summarize the definition of rates in Fig. 2B and their derivation in the SI. Thanks to the mechanistic derivation of the rate equations, we were able to constrain most model parameters using literature values for *E. coli* (see SI §S2.2). Only three parameters—the rate of damage to healthy ribosomes, $k_{dam}$, as well as the maximal transcription rates, $\omega_r$ and $\omega_{ba}$—remained unconstrained and formed the basis of the analysis that follows below.

The model of Rtc-regulated RNA repair describes how Rtc expression in *E. coli* adapts to damage to rRNA or tRNA and the subsequent actions of the Rtc proteins in maintaining translational RNA (Fig. 2A). It predicts the level of healthy rRNA or tRNA available to cells, based on their ability to induce Rtc expression to counter the imposed damage, and thus, the cellular capacity to sustain translational activity. We next set out to characterise the dynamic response landscape of Rtc expression and its consequences for the translational capacity of cells.

## Robust bistability of the Rtc model suggests heterogeneous levels of growth rescue

Activation of the Rtc system is characterised by expression of the RtcBA proteins. We first investigated how Rtc expression adapts to the rate of damage imposed on cells. To this end, we performed a stability analysis of the model to determine the steady-state levels of Rtc and translational RNA species at various damage rates (see Methods).

We consider the absence of damage, i.e. $k_{dam} = 0$, representative of wild-type *E. coli* where Rtc is not active. When damage rates increase, for example through the addition of a ribosome-targeting antibiotic, the model predicts an initial increase in steady-state RtcBA protein expression due to an adequate availability of healthy ribosomes (Figs. 3A and S2A). Preceding translation, transcription of *rtcBA* (Figs. S3C, S4C) occurs due to an increase in tagged RNA causing RtcR activation (Fig. 3A, S2A).

Steady-state expression of RtcBA is predicted to peak when activation levels of RtcR* saturate (Fig. 3A). When damage rates increase further, steady-state RtcBA expression decreases as the levels of healthy ribosomes available for translation decline. Eventually, expression is predicted to collapse at high rates of damage, when there is no longer sufficient healthy RNA to sustain translation of the Rtc repair proteins. In the collapsed state there is a build up of damaged RNA due to a lack of sufficient RtcBA to carry out repair. RtcR expression, which was assumed to be constitutive, is less affected by the damage rate and follows declining levels of healthy RNA (Figs. 3A, S2A).

Interestingly, the stability analysis revealed that the Rtc model displays bistability across a range of damage rates. Bistability of a dynamical system denotes the co-existence of two stable steady-states for a given condition. When damage rates decrease from initially high

levels (rates above the grey range in Fig. 3A), the system remains in the collapsed state even for damage rates (rates within the grey range) that allow Rtc expression in cells which previously experienced low damage (rates below the grey range). This phenomenon, whereby the system adopts one stable state over the other based upon a memory of previous conditions (in this case damage), is called hysteresis. It is a key feature of bistability and typically arises as a result of positive feedback, especially in combination with ultrasensitivity[46], both of which are present in the Rtc system. Ultrasensitivity is exhibited through the cooperative activation of RtcR by tagged RNA, and positive feedback is exerted in the model at two stages: 1) through RtcA which is required to activate its own expression, and 2) through healthy RNA necessary for translation of the repair proteins.

The presence of bistability suggests that two distinct Rtc expression states can co-exist for a range of damage inflicted on cells for Rtc proteins and ribosome species (Fig. 3A). These cell states represent an Rtc-on state, where there is expression of *rtcBA* and repair of translational RNA, corresponding to resistant cells, and an Rtc-off state with insufficient healthy RNA to sustain protein expression, corresponding to susceptible cells. At damage rates within the bistable region (grey range in Fig. 3A), the system will enter the on-state if previous exposure to damage has been lower, and enter the off-state if previous exposure has been higher than levels within the bistable region. The model therefore suggests that for a range of damage conditions the structure of Rtc regulation promotes phenotypic heterogeneity, where resistant and susceptible cells can co-exist within a form of hetero-resistant cell population.

We next aimed to identify whether bistability was robust across a range of parameter conditions, or if it was specific to a small subset of conditions. We analysed the sensitivity of model predictions to variation in key parameters (see Methods), including the unknown maximal transcription rates of *rtc* genes, $\omega_{ba}$ and $\omega_r$, as well as the dilution rate $\lambda$ and cellular ATP levels, which we could only loosely constrain within a physiological range due to their inherent variation.

Sensitivity analysis suggests that the Rtc system displays bistability robustly throughout a range of parameter conditions (Fig. 3B, C and S2B-C). Occurrence of bistability at any damage rate within the range we considered (between 0 and 0.8 min⁻¹) was largely insensitive to ATP concentrations, however, low ATP concentrations affected the dilution rates that support bistability. Dilution rates that support bistability were further predicted to depend on inducibility of the *rtc* repair genes, where higher inducibility increases the range of dilution conditions that support bistable responses (Fig. 3B).

We further analysed how Rtc expression levels and the region of bistability, i.e. damage rates where resistant and susceptible cells are predicted to co-exist, depend on individual changes in parameter values. Figures 3C and S2C show the effect of cellular ATP concentration (top), dilution rate (middle), and maximal transcription rate of the *rtcBA* genes $\omega_{ba}$ (lower panel) on RtcB expression. Changes in the transcription rate of *rtcR* had qualitatively the same effect as changes in *rtcBA* transcription (Figs. S3B, S4B).

Higher ATP concentrations support higher overall expression in the Rtc-on state and widen the region of damage rates where bistability is displayed. The widening of the bistable region at high ATP levels largely resulted from an increase of its upper boundary, whereas the effect on its lower boundary was minor, explaining why bistability occurred at all ATP concentrations in Fig. 3B. Conversely, lower dilution rates resulted in higher Rtc-on expression and regions of bistability at higher damage rates. For low dilution rates (grey line), the region of bistability was beyond damage rates of interest (Fig. S3B). The system thus robustly displays bistability across the range of physiologically realistic dilution rates, however, not necessarily within the range of damage rates of interest (cf. Fig. 3B). Increasing the maximal transcription rates of the *rtc* genes, $\omega_{ba}$ and $\omega_r$, represents an increase in Rtc inducibility and has qualitatively similar effects on bistability as

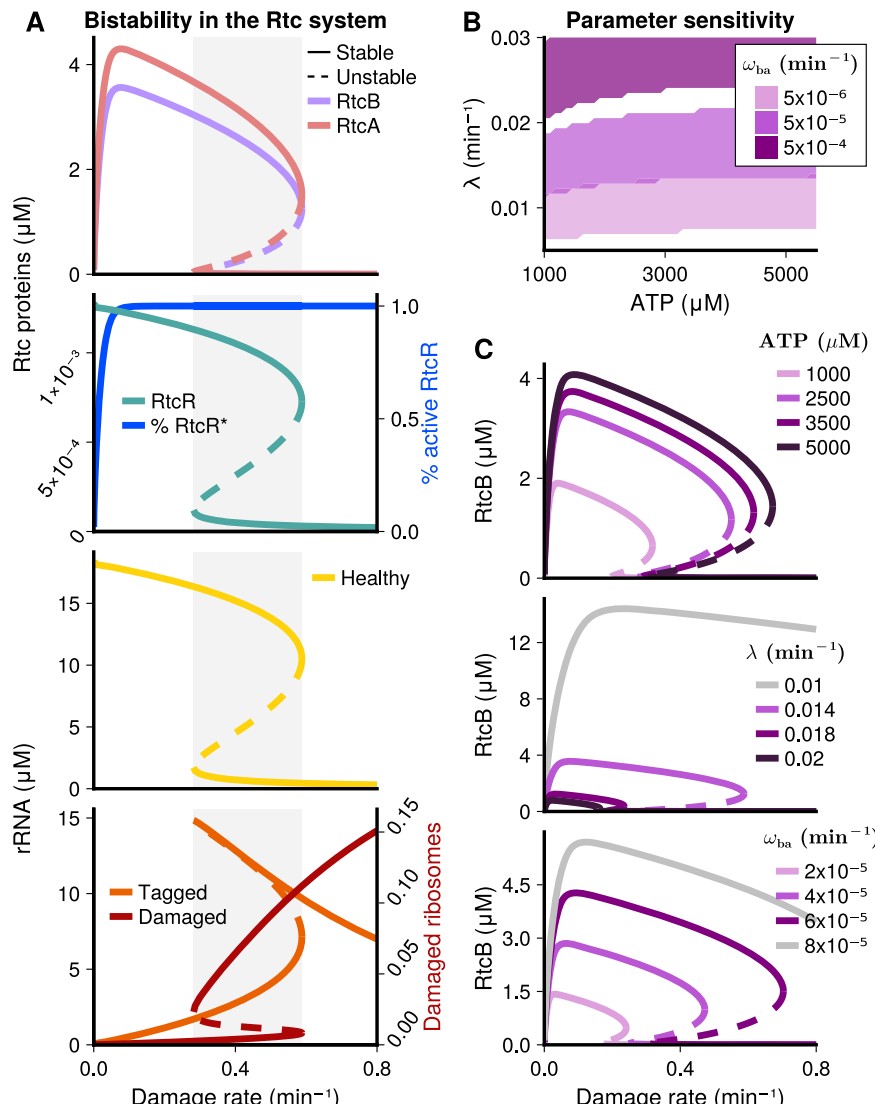

**Fig. 3 | Steady-state responses of the rRNA-repair model. A** Stability analysis: Bifurcation diagrams for the Rtc proteins (top) and rRNA species (bottom) show a range of damage rates (grey region) in which two stable cell states (solid lines) and one unstable steady-state (dashed lines) co-exist. If damage increases from initially low levels (below grey region), cells are predicted to enter the 'Rtc-on', *resistant* state (upper solid branch for all species but damaged and tagged rRNA, lower solid branch for latter) that enables rRNA repair, and consequently, maintenance of sufficiently high levels of healthy rRNA to sustain translational activity. Conversely, if damage decreases from initially high levels (above grey region), cells are predicted to remain in an 'Rtc-off' state (lower solid branch except for damaged and tagged rRNA), where RNA damage levels exceed the capacity of repair and prevent translation of Rtc proteins. **B** The system displays bistability across a range of parameter combinations. Purple regions represent combinations of ATP concentrations and dilution rates, that display bistability within the range of damage

rates considered (0 to 0.8 min⁻¹). Shades of purple indicate how the regions depend on the inducibility, i.e. the maximal transcription rate $\omega_{ba}$, of the *rtcBA* genes, where higher inducibility supports bistability across a wider and increasing range of dilution conditions. **C** Parameter sensitivity: ATP-availability (top), dilution rate (middle) and maximal *rtcBA* transcription rate (bottom) impact the magnitude of Rtc expression and the region of bistability. Higher ATP concentrations and *rtcBA* transcription yield higher expression states, whereas dilution lowers expression. The region of bistability widens for higher ATP concentrations, and it shifts towards higher ranges of damage rates for lower rates of dilution and higher transcription. Grey lines indicate conditions where the region of bistability has shifted beyond damage rates of interest, and so the system is monostable within the considered range of damage. Unless stated otherwise, parameter values are given in Table S1. Source data are provided in the source data file.

decreasing dilution. To account for the scenario where RtcA acts on 2'-phosphate ends, albeit at a slower rate (0.1 min⁻¹) to 3'-phosphate termini, we decreased the $k_{tag}$ parameter (SI Fig. S5). We observe that at the slower rate of tagging of 2'-phosphate termini for both rRNA and tRNA repair the model displays bistability at extreme parameter values and is less robust than bistability observed in 3'-phosphate repair.

We also performed a global sensitivity analysis, whereby we varied 15 different parameters according to double of their reported standard deviations (see §2.3). Bistability was robust for both models, out of 5 million combinations of varied parameter values, more than 95% of them gave a bistable result for the rRNA model (Figs. S6 and S7).

Finally, we asked in how far the presence of bistability depended on assumptions of static dilution ($\lambda$) and the rate of ribosome influx ($k_{in}$), which are known to correlate with ribosome levels (see SI §1.4.2 and Fig. S8), as well as on the assumption that RtcR acts as a hexamer (Figs. S3D, S4D), which was assumed based on knowledge of other bEBPs, neither of which limited predictions on bistability.

To conclude, the model of Rtc-regulated RNA repair displays bistability, suggesting that isogenic cells can adopt two distinct response phenotypes when faced with the same level of damage inflicted by translation-targeting antibiotics. These phenotypes correspond to an Rtc-off state, where cells are translationally inactive, i.e.

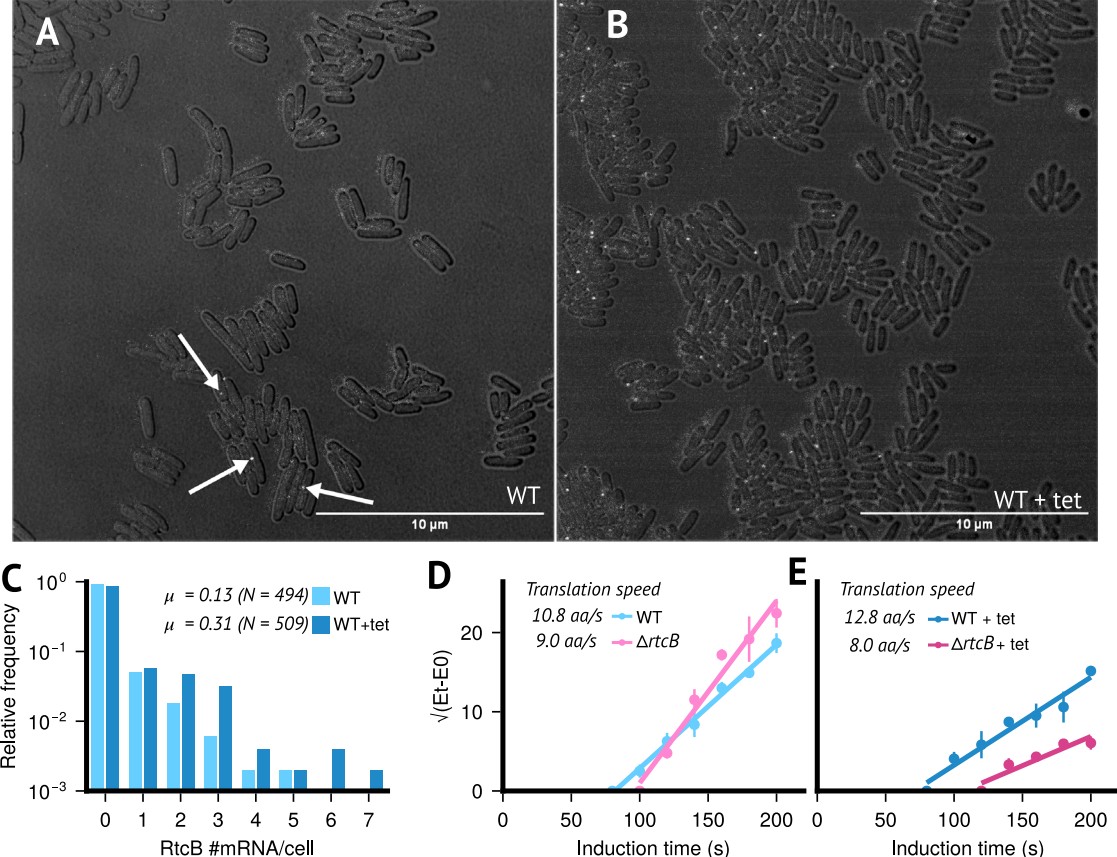

**Fig. 4 | RtcB is involved in the cellular response of *E. coli* bacteria to sub-inhibitory concentrations of Tetracycline.** Brightfield and 6-TAMRA channel overlay images of *E. coli* BW25113 wild-type cells at mid-exponential phase. **A** Untreated and (**B**) treated cells exposed to 1.5 µg/ml tetracycline for 1 hour. Images were acquired using a Leica Stellaris 8 confocal microscope, and overlays of brightfield and 6-TAMRA channels were generated using ImageJ's overlay function. Fluorescent signals detected in the 6-TAMRA channel appear as white spots, corresponding to *rtcB*-specific fluorescent DNA probes hybridised to *rtcB* mRNA molecules. Images from the 6-TAMRA channel were analysed using Spätzcells software to determine the total number of *rtcB* mRNA molecules per cell. **C** The output data from Spätzcells were used to calculate the mean (µ) and relative frequency of *rtcB* mRNAs per cell in the population (nWT = 494 and nWT+tet = 509, experiments were performed with three independent biological replicates). **D**, **E** The impact of sub-inhibitory concentrations of tetracycline on the speed of translation elongation in presence and absence of *rtcB* was measured in vivo using FusA-LacZα reporter assays[67]. Shown is the square root of enzymatic activity over induction time. **D** Untreated and **E** treated cells exposed to 1.5 µg/ml tetracycline for 1 hour. Data points represent the mean and standard deviation of 3 biological replicates. The *x*-intercept indicates the time it takes to translate FusA-LacZα. Source data are provided in the source data file.

susceptible to the inflicted damage, and an Rtc-on state, where cells maintain translational activity and thus exhibit a form of adaptive and transient resistance to the assault. Bistability is displayed robustly across the range of physiologically feasible parameter values, indicating that structural aspects of Rtc-regulation inherently promote this phenotypic heterogeneity.

### Rtc expression displays heterogeneity and impacts translational capacity

To test for heterogeneity, a granular characterisation of *rtcBA* expression is required, which to our knowledge has not been undertaken so far. We therefore measured expression across single *E. coli* cells using single-molecule RNA-FISH, monitoring *rtcB* mRNA copy numbers in the absence and presence of the ribosome-targeting tetracycline, which has previously been shown to induce *rtcBA* expression[10] (see Figs. 4A-C and S9 and Methods).

Consistent with previous data[10], induction increased mean expression (µ in Fig. 4C) by approximately 2.5-fold. Crucially, the distributions of mRNA copy numbers per cell displayed a large majority of non-expressing cells in both uninduced and induced conditions, implying that average induction was driven by only a small proportion of expressing cells (Fig. 4C). Our data indicate that average induction,

as reported previously[10,11], is a result of (a) higher copy numbers in expressing cells (one-sided *z*-test, *p*-value < 10⁻³) and (b) a near doubling of cells expressing *rtcB* (22% vs 13%, see Methods). Overall, the expression data validate that induction at the population-level is modulated via heterogeneity across individual cells, both, in terms of higher expression levels and a higher proportion of expressing or 'on' cells.

To moreover characterise the impact *rtcB* expression exerts on translational capacity, we quantified in vivo translation-elongation speed using a LacZα-complementation system (see Methods). Comparing translation speed in wild-type and Δ*rtcB* cells, consistent with earlier data[11], we observed that lack of *rtcB* reduced translation-elongation speed even without addition of tetracycline (Fig. 4D), suggesting that a baseline level of damage may already be prevalent without drug-exposure. Reduction of translation-elongation speed was however more pronounced when cells were exposed to tetracycline (Fig. 4E), implying that *rtcB*-expressing cells can sustain higher translational capacity upon exposure to the drug.

As predicted by our model, the combined data indicate that *rtcBA* expression improves translational capacity amid challenge from a ribosome-targeting antibiotic and that expression displays considerable heterogeneity across single bacterial cells. Together, our

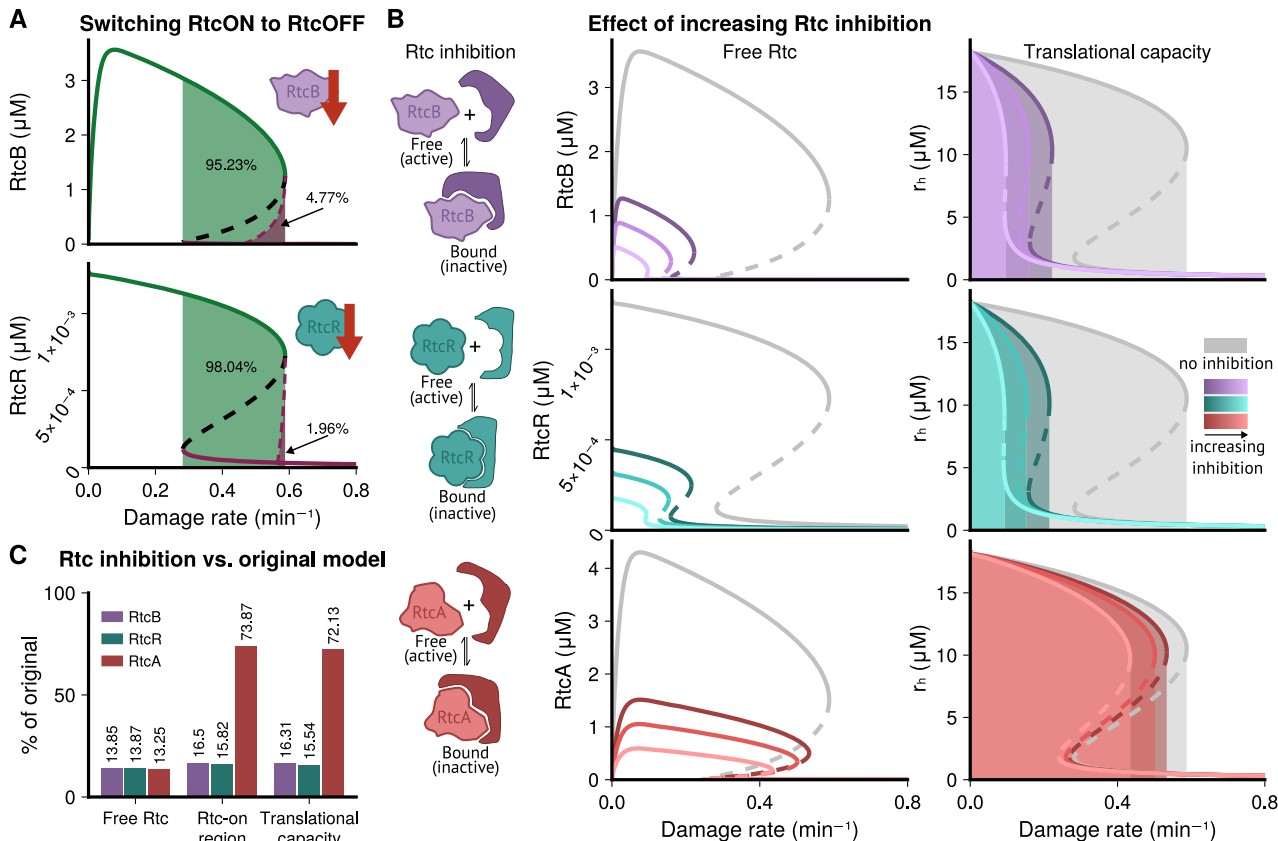

**Fig. 5 | Inhibiting Rtc proteins is predicted to reduce levels of resistance.** Predictions for assumed damage and repair of rRNA; equivalent analysis of tRNA damage and repair is shown in Fig. S10. **A** When perturbing initial conditions of one species at a time, only perturbations to RtcB (top) and RtcR (bottom) from their 'on' steady states, can 'switch off' Rtc expression. Green areas mark initial values that are insufficient to induce a switch, and purple areas values that drive the system to switch to the susceptible Rtc-'off' state. For both species, switching is only possible for high damage rates within the bistable region. RtcB can cause a switch for a wider range of damage rates. Only perturbations of RtcR within the top section of the bistable region can produce a switch to the 'off' state. **B** Inhibiting Rtc proteins decreases predicted concentrations of free Rtc (left). Inhibiting RtcB (top) or RtcR (middle) reduces the range of damage conditions where Rtc is expressed and resistance is possible. Inhibiting RtcA decreases the concentration of free RtcA but barely changes the range of inducing damage conditions (bottom). Levels of healthy ribosomes (right), which determine the translational capacity of cells, display substantial reduction upon inhibition of both RtcB and RtcR but not upon inhibition of RtcA. **C** Summary of Rtc inhibition comparing the effects of the strongest level of inhibition (see Table S2) when targeting RtcB, RtcR or RtcA to outcomes without inhibition. Free Rtc concentration was obtained by calculating the average percentage difference for all values of damage across the curve. Targeting any Rtc protein has a similar effect on free protein concentrations. Rtc-on/ resistant cell state metrics were obtained by calculating the percentage difference between the size of the Rtc-on region with and without inhibition. The range of damage conditions that enable the Rtc-on/resistant cell state is substantially reduced when inhibiting RtcB and RtcR but not RtcA. Similarly, inhibiting RtcA has minor effects on translational capacity compared to RtcB and RtcR. We quantify translational capacity by the shaded area under the on-curve of healthy ribosomes in (**B**) relative to that without inhibition. Source data are provided in the source data file.

computational and experimental findings suggest that the *E. coli rtcBA* response to ribosome-targeting antibiotics may be underpinned by a form of hetero-resistance.

**Targeted inhibition of Rtc may reduce Rtc-induced resistance**
We next asked if the system could be manipulated such that *rtcBA*-expressing, or resistant, cells switch to the susceptible state. Since initial conditions determine which stable steady state is adopted, we started by analysing the sensitivity of model outcomes to perturbations in initial values by systematically perturbing one variable at a time.

Figure 5A (and Fig. S10 for corresponding tRNA analysis) shows the effect of perturbing initial concentrations of the RtcB and RtcR proteins away from their stable 'on' levels. Switching steady states requires perturbations beyond the values at the unstable steady state, because we only perturb initial concentrations of one variable at a time. Perturbing initial concentrations of RtcB was most effective at 'switching off' Rtc expression, and thus, at rendering the system

susceptible. At high damage rates within the bistable region, the switch only requires a small decrease from the steady-state RtcB level to switch expression off. For lower damage rates, however, switching requires a more substantial decrease in RtcB levels, and for damage below a certain rate, the system was locked in the Rtc-on state, i.e. perturbation of RtcB alone was insufficient to switch expression off. Perturbation of initial RtcR concentration was also able to switch expression off, but only within a small range of high-damage conditions. We also investigated the impact of perturbations in initial RtcA concentrations but found that, regardless of the magnitude of perturbations, they were in almost all cases insufficient to switch Rtc expression off in rRNA repair and completely insufficient in tRNA repair (see Figs. S3E and S4E).

Driven by the results of the above sensitivity analysis, we wondered whether inhibition of RtcB might be effective at rendering cells more susceptible to damage. Here, we hypothesise that an inhibitor could be used in combination with an antibiotic to successfully treat bacterial infections without Rtc induction and therefore without Rtc-

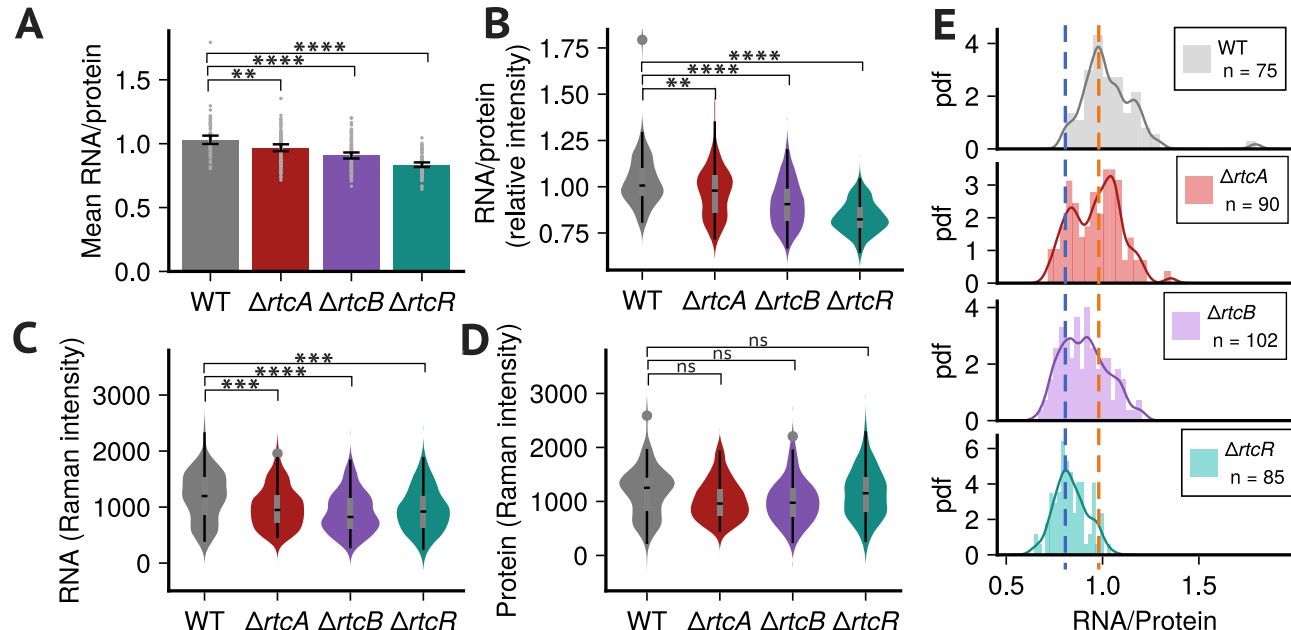

**Fig. 6 | Deletion of individual *rtc* genes decreases ribosome levels in tetracycline-treated cells. A–D** Comparison of single-cell Raman intensities from knockout strains (Δ*rtcA*, n = 90; Δ*rtcB*, n = 102; Δ*rtcR*, n = 85) to WT (n = 75). All experiments were performed with three independent biological replicates. Asterisks denote significance with *: < 5%, **: < 1%, ***: < 0.1%, ****: < 0.01%, ns: not significant. **A** Mean RNA/protein ratios (bars), 95% confidence intervals (error bars) and single-cell ratios (dots). Mean ratios are significantly lower than WT for each knockout strain (WT: 1.03 ∈ [1.00, 1.06], Δ*rtcA*: 0.97 ∈ [0.94, 1.00], Δ*rtcB*: 0.91 ∈ [0.88, 0.93], Δ*rtcR*: 0.84 ∈ [0.82, 0.85]), with reduction highly significant in Δ*rtcB* and Δ*rtcR* (unpaired two-sided t-tests for each knockout strain compared to the WT; Δ*rtcA* t-statistic: 2.88, degrees of freedom (df): 150, p-value: 0.0045; Δ*rtcB* t-statistic: 5.98, df: 143, p-value: $1.7 \cdot 10^{-8}$; Δ*rtcR* t-statistic: 10.25, df: 116, p-value: $5.8 \cdot 10^{-18}$). **B–D** Box plots highlight median, interquartile range (IQR), with whiskers spanning third and first quartiles ± 1.5 × IQR. **B** Distributions of RNA/protein ratios

indicate that all knockout strains display significantly reduced ribosome concentrations compared to WT, with levels in Δ*rtcB* and Δ*rtcR* lower than those in Δ*rtcA*. **C** Distributions of total RNA and **D** total protein demonstrate that deletions have no significant effect on total protein concentrations but reduce total RNA. For (**B–D**) asterisks indicate significance levels in a one-sided Mann Whitney U test comparing knockout levels to WT. For RNA/protein ratios (**B**) (p-values Δ*rtcA*: 0.008, Δ*rtcB*: $7.9 \cdot 10^{-9}$, Δ*rtcR*: $3.7 \cdot 10^{-19}$) and total RNA (**C**) (p-values Δ*rtcA*: $8 \cdot 10^{-4}$, Δ*rtcB*: $4.1 \cdot 10^{-6}$, Δ*rtcR*: $2 \cdot 10^{-4}$), we tested if knockout levels were smaller than WT, and for total protein (**D**) (p-values Δ*rtcA*: 0.991, Δ*rtcB*: 0.997, Δ*rtcR*: 0.736) if levels were greater than WT. **E** Histograms for each strain are shown with a kernel density estimate overlayed (solid lines). The distribution of the Δ*rtcA* strain displays two distinct modes that coincide with the modes of higher ribosome levels in WT (orange dashed line) and those of lower levels, i.e. more susceptible, in Δ*rtcB* and Δ*rtcR* (blue dashed line). Source data are provided in the source data file.

induced resistance (Fig. 1C). To this end, we introduced a new parameter in the model, representing an inhibitor that can reversibly bind RtcB and prevent its sealing action to repair ribosomes (see §S1.4.1). Figure 5B (top) shows that RtcB inhibition is predicted not only to cause a reduction in the concentration of free (unbound) RtcB but also a considerable reduction in the range of damage conditions that induce Rtc expression. Effects are amplified the stronger the inhibitor binds RtcB (see SI Table S2 for inhibition parameters). Notably, reduction of free RtcB is accompanied by a substantial reduction in translational capacity, in terms of levels of healthy RNA, suggesting reduced resistance to damage.

We also tested inhibition of the other two Rtc proteins, RtcR and RtcA. The effects of inhibiting RtcR were comparable to those of inhibiting RtcB in terms of reducing Rtc induction and the range of inducing damage conditions, as was the effect on translational capacity (Fig. 5B, middle). Inhibiting RtcA, however, had a much weaker effect. While levels of free RtcA were reduced, similar to those of RtcB when RtcB was the target of inhibition, the range of inducing damage conditions was only reduced a fraction of the amount compared to that of RtcB and RtcR (Fig. 5B, bottom). Importantly, the effect on translational capacity was also minor, suggesting that inhibition of RtcA would be ineffective at reducing levels of resistance. Figure 5C summarizes predicted effects of inhibiting the different Rtc proteins on suppressing Rtc induction and associated resistance in terms of rescued translational capacity. Our analysis suggests that both RtcB and RtcR, but not RtcA, may be potent targets to prevent induction of *rtcBA* and its induced resistance to translation-targeting antibiotics.

We then went on to seek experimental validation of our computational predictions. Since no Rtc inhibitors are currently known, we characterised translational capacity when expression of individual *rtc* genes is altered in cells that are exposed to a ribosome-targeting antibiotic. Using Raman microspectroscopy[47] (see Methods), we measured ribosome concentrations, by means of RNA/protein ratios, in wild-type *E. coli* and individual *rtc* knockout strains (Δ*rtcA*, Δ*rtcB* and Δ*rtcR*) at single-cell resolution (see Methods).

In the presence of tetracycline, all *rtc* knockout strains exhibited lower mean RNA/protein ratios than wild-type cells (Fig. 6A). The data further revealed that, compared to wild-type, distributions across single cells display significantly lower RNA/protein ratios in Δ*rtcB* and, even more so, in Δ*rtcR* cells (Fig. 6B, see Methods). While Δ*rtcA* cells also showed significantly lower RNA/protein ratios than wild-type cells, differences were not as highly significant as in the Δ*rtcR* and Δ*rtcB* strains. We further observed that lower RNA/protein ratios were due to lower total RNA levels rather than increased total protein levels, where differences were non-significant (Fig. 6C, D).

Distributions of Δ*rtcR* and Δ*rtcB* RNA/protein ratios displayed distinctly different modes from the wild-type strain (Fig. 6E). In contrast, Δ*rtcA* cells displayed a bimodal distribution, where modes coincide with those of the wild-type and the other knockout strains (Fig. 6E, orange vs blue dashed lines). Consistent with our model predictions, the latter suggests that lack of *rtcA* leads to a heterogeneous response where some cells are able to maintain full translational capacity, similar to the wild-type, but some cells display impaired translational capacity, similar to that of Δ*rtcB* and Δ*rtcR*.

Consistent with our model predictions, our data therefore suggest that cells lacking *rtcR* and *rtcB* maintain lower translational capacity, presumably due to impaired repair action, and deletion of *rtcA* is not as potent in locking cells in a state of lower translational capacity. Equivalent experiments performed in the absence of tetracycline display no significant differences in distributions between wild-type and all three *rtc* deletion strains, indicating that this is a stress-specific effect of the Rtc system induced by the antibiotic (Fig. S11).

Our results motivate further characterisation of the Rtc system, in particular of RtcB and RtcR, to identify suitable inhibitors and test their efficacy in rendering bacterial cells susceptible to ribosome-targeting antibiotics. Given the wide use of these antibiotics and strong conservation of the *rtcBA* system across several clinically relevant pathogens, our results may point to a vigorous route for potentiating treatments and reducing levels of resistance.

## Discussion

We developed a computational model of Rtc-induced maintenance of long-lived translational RNAs in *E. coli*. The model incorporates key aspects of the current body of knowledge around *rtcBA* regulation and its role in invoking adaptive resistance to translation-targeting antibiotics. It considers transcription and translation of the three Rtc proteins and their mechanisms of action to aid research into determining the role of Rtc in antibiotic resistance.

We made various assumptions to keep the model simple yet realistic. Based on information available about other bEBPs, we assumed that RtcR is constitutively expressed and that negative auto-regulation of RtcR prevents oligomerisation of its subunits. As RtcR has been shown to form apparent hexamers, similar to other bEBPs[11,43], we assume cooperative binding of the activating ligand, which based on current evidence[19], we assumed to be RNA ends that have been cyclysed by RtcA. We model all energy-dependent reactions using standard mass action and Michaelis-Menten kinetics. Although RtcA and RtcB are dependent on different NTPs, we consider ATP in the model as a general measure of cellular energy, hence it accounts for other NTPs in addition to ATP. We assumed that enzyme reactions for the mechanisms of action of RtcA, RtcB and RtcR are in quasi-steady state allowing derivation of net rates for healing, sealing and transcription initiation, respectively. Given the affinities for ATP of both RtcA and RtcB we assumed their repair reactions are saturated with ATP (see SI)[7,48]. We further assumed that ribosome-targeting antibiotics, or oxidative stress inducing agents, indirectly damage rRNA or tRNA. Finally, we assumed that the rate of influx of healthy ribosomes or tRNA is constant, same as the rate by which all model variables are diluted due to growth.

Based on previous findings that the *E. coli rtcBA* system is induced upon both ribosome-targeting antibiotics as well as agents that cause oxidative damage, we assumed that damage and repair may involve rRNA within otherwise intact ribosomes or tRNA. Whilst rRNA and tRNA affect translation in different ways, we did not observe qualitative differences in the predicted phenotypes following Rtc-regulated maintenance of rRNA vs tRNA, or its inhibition (see also SI §S1.3). We also analysed how uncertainty surrounding several of the other assumptions may limit predictions and found that none of those investigated substantially altered qualitative predictions and the conclusions drawn from them (see SI).

Our model analysis suggests that the Rtc system displays bistability, i.e. that distinct phenotypes representing Rtc-off or susceptible and Rtc-on or resistant cell states can stably co-exist, and that bistability is displayed robustly across a wide range of physiologically plausible parameter combinations. Bistability has been linked to antibiotic tolerance and persistence, where isogenic cells can adopt one of two stable steady-states and switch between them, as has been observed in expression of virulence genes in *Salmonella* and in *E. coli*'s central carbon metabolism[49,50]. Our computational results

predict that, through bistability, Rtc expression may be underpinned by considerable levels of cell-to-cell heterogeneity.

The model predictions motivated an exploration of *rtcBA* expression at the single-cell level. Our experiments provided evidence that modulation of *rtcBA* expression is indeed underpinned by cell-to-cell heterogeneity, and they confirmed that expression benefits cells through improved translational capacity upon exposure to a ribosome-targeting antibiotic. Our results therefore suggest that induced levels of resistance may in fact form a type of hetero-resistance. These findings raise important questions into the specific conditions that trigger heterogeneity in antibiotic responses, especially given that other studies, under different experimental conditions, have previously not suggested bistable or heterogeneous responses to the *rtcBA*-inducing antibiotics tetracyline and chloramphenicol[27,51,52]. In view of alarming levels of antibiotic resistance in *E. coli* and other clinically relevant species that carry the *rtcBA* operon, future work will be required to characterise this form of resistance to commonly used antibiotics.

Insights into the mechanisms promoting heterogeneity of Rtc expression allowed us to explore potential molecular targets aimed at suppressing Rtc expression and rendering cells more susceptible to translational assaults by antibiotics. We computationally screened the Rtc model for molecular targets which revealed that RtcB or RtcR would be the preferred targets for Rtc inhibition and that inhibition of RtcA was not predicted to achieve the same level of resistance suppression. Comparing the predicted efficacies of inhibiting RtcA vs RtcB or RtcR, we hypothesize that differences may result from their complementary roles in exerting positive feedback onto RtcBA expression. RtcB directly affects the availability of functional (ribosomal or transfer) RNAs that are essential for translation, and RtcR controls the expression of RtcB, potentially explaining why their inhibition has similar effects in terms of RtcBA induction. RtcA, in turn, does not directly affect the pool of healthy RNAs, and its inhibition, despite impacting the Rtc-inducing ligand, appears insufficient to render cells susceptible to damage to the same extent as RtcB and RtcR. Furthermore, RtcA processes damaged RNAs at a considerably faster rate than RtcB, suggesting that the sealing action by RtcB may constitute a rate-limiting step in RNA repair.

Inspired by these predictions, we sought validation through quantifying single-cell ribosome contents across wild-type and individual *rtc* knockout strains. When exposed to tetracycline, the knockout strains displayed lower ribosome levels than wild-type cells, with Δ*rtcA* displaying a lesser effect on the translational capacity than both Δ*rtcB* or Δ*rtcR*. The same effect was not observed in untreated cells, therefore, highlighting that the Rtc system rescues translational capacity of cells in stress-inducing conditions but not in non-stress conditions. While gene deletion presents a different cellular intervention than inhibition, our data nevertheless provide strong initial validation for the in silico predictions which favoured targeting RtcB or RtcR over RtcA to suppress translational recovery.

Taken together, our computational and experimental results suggest targeting RtcB or RtcR with an inhibitor in combination with translation-targeting antibiotics as a promising strategy to suppress the activation of Rtc and alleviate levels of resistance seen upon exposure. Whilst these targets provide convincing results in silico via protein inhibition, and in vivo through *rtc* deletion, we recognise the difficulties associated with identifying effective inhibitors and targeting RtcR and RtcB in vivo. Firstly, as a regulator of gene expression, RtcR may be more prone to resistance mutations[53–55], which could enable bacteria to counteract the inhibition of RtcR. Secondly, the highly conserved nature of the Rtc system means that RtcB is not just an essential protein for bacterial pathogens but also for their host organisms. Its inhibition in combination with an antibiotic could thus have adverse affects on the infected organism, and an ideal inhibitor may therefore be required to target regions of RtcB that make the

inhibition unique to *E. coli* and closely related species carrying the *rtcBA* operon. Thirdly, some bacteria, including several *E. coli* strains encode more than one RtcB ligase[12]. While some of the homologs have been shown to repair complementary RNA substrates[12], we cannot exclude that there may be redundancy between them, which may under certain stresses limit the efficacy of inhibition. In view of these caveats, our results motivate a host of experiments to assess the feasibility of targeting components of the Rtc system, the clinical relevance of which was also recently highlighted by work that identified a role of RtcB in protecting *S. enterica* from oxidative host-immune assault[13].

We presented a model-guided approach that provided insights and explanation into previously observed phenomena around Rtc-regulated RNA repair. Our work uncovered how known aspects of Rtc regulation have previously unknown and potentially decisive implications for the nature of resistance conferred by Rtc, and it pinpointed potential strategies to alleviate it. Possible limitations include the static nature of some of the physiological model parameters that are known to vary over time, which we addressed partially. Future work may build on our approach to investigate the role of a dynamic and adaptive cell physiology in shaping the adaptive resistance conferred by Rtc. Further, our data on single-cell *rtcB* expression provided evidence of cell-to-cell heterogeneity, but not whether the cause of heterogeneity is indeed bistability, which would require observing hysteresis. We note that other factors also impact heterogeneity, for example, given the low levels of *rtc* expression we observed, stochasticity is also likely to contribute, raising interesting questions for future investigation. RNA repair is an area of research that holds crucial insights into the maintenance of living cells. Our work advocates a systemic grasp of the physiological role of RNA repair via Rtc, using computational models to test competing hypotheses and generate fresh ones, and so it provides rigorous tools to accelerate future discoveries.

## Methods

The work only included non-pathogenic, non-infectious *E. coli* model bacteria. No animals or humans were involved in this study. Research with this model bacterium was approved by our internal health and safety committee.

### Model simulations

The model has been coded in Julia v1.11 using the ModelingToolkit.jl (v9.54.0) package[56]. Full details of the model, including all equations and derivations, can be found in the SI. Code for defining and solving models can be found at https://doi.org/10.5281/zenodo.17048324[57]. We used the Julia package DifferentialEquations.jl (v7.15.0)[58] for all model simulations and simulated the model using the Rodas4 solver with default tolerances.

### Stability analysis

BifurcationKit.jl (v0.4.4)[59] was used to perform the stability analysis. Some parameters that were important in solving the system with the package include 'ds' and 'dsmax' in the ContinuationPar settings. Finally, in the continuation function, the parameter '$\theta$' was also very sensitive to solving the model correctly. The required parameters for the bifurcation kit in Julia that were used to solve the model can be found in the provided code. Some other Julia packages were used for plotting, including CairoMakie.jl (v0.12.18)[60], DataFrames.jl (v1.7.0)[61] and Interpolations.jl (v0.15.1)[62].

### Bacterial strains and growth conditions for single-molecule RNA FISH and in vivo translation elongation speed.

Experiments were performed with *Escherichia coli* K-12 BW25113 strains. The *rtcB* gene deletion mutant (JW3384) was from the KEIO collection (Horizon Discovery) in which the *rtcB* open reading frame was replaced with a kanamycin cassette[63]. For single molecule RNA FISH and translation elongation speed measurements, cells were sub-cultured into fresh Luria-Bertani (LB) broth at 37 °C from an LB overnight culture to a starting $OD_{600}$ of 0.1 and harvested at 0.5 in a volume of 15 ml in a 50 ml flask. Tetracycline treated cells were exposed to 1.5 µg/ml tetracycline for 1 hour.

### Single-molecule RNA FISH

After culturing the cells as described above, they were resuspended and fixed in 1 ml of ice-cold 1 × PBS in DEPC-treated water containing 3.7% (v/v) formaldehyde and incubated for 30 minutes at room temperature. The cells were washed twice with 1 ml of 1 × PBS in DEPC-treated water, then resuspended in 1 ml of 70% (v/v) ethanol in DEPC-treated water and incubated for 1 hour at room temperature to permeabilise. The cells were washed with 1 ml of 2 × SSC in DEPC-treated water containing 40% (w/v) formamide and incubated overnight at 30 °C with hybridisation buffer (2 × SSC in DEPC-treated water, 40% (w/v) formamide, 10% (w/v) dextran sulfate, 2 mM ribonucleoside-vanadyl complex, and 1 mg/ml *E. coli* tRNA) and 1 µM 6-TAMRA-labelled Stellaris™ RNA FISH probes complementary to the *rtcB* gene sequence. 6-carboxytetramethylrhodamine succinimidyl ester (6-TAMRA) labelled DNA probes against *rtcB* were purchased from LGC Biosearch Technology and designed using the Stellaris® Probe Designer (full details in Supplementary Table S5). The length of each oligo was 20 nt with a minimal spacing of 2 nt between them and a masking level of 1-2. After hybridisation, 10 µl of the cells were washed twice in 200 µl of ice-cold wash solution (40% (w/v) formamide and 2 × SSC in DEPC-treated water) and incubated for 30 minutes at 30 °C. The chromosomal DNA was stained with DAPI-containing wash solution (40% (w/v) formamide, 2 × SSC in DEPC-treated water, and 10 µg/ml DAPI) for 30 minutes at 30 °C. The cells were resuspended in 10 µl of 2 × SSC in DEPC-treated water, from which 2 µl was spotted onto the centre of a 1% (w/v) agarose gel. Once dry, a 1 × 1 cm square was cut around the sample and transferred to a 76 × 26 mm microscope glass slide for cell imaging.

### Quantification of *rtcB* mRNA molecules.

Cells were imaged using a Leica Stellaris 8 confocal microscope to quantify *rtcB* mRNA molecules using smRNA FISH[64]. Five z-slices at 200 nm intervals were acquired for each channel (brightfield, DAPI, and 6-TAMRA) across multiple x/y stage positions. The resulting 16-bit .tif images from all three channels were used to generate cell segmentations using the Schnitzcells software[65] in MATLAB (MathWorks). The *rtcB* mRNA copy numbers in single cells were quantified with the Spätzcells program[64] in MATLAB, using the 6-TAMRA channel images and the cell segmentations. Fluorescent spots within the segmented cells were detected and differentiated from nonspecific background signals by setting a false-positive threshold using Δ*rtcB* cells as a negative control (Fig. S9). False-positive spots were excluded by setting the threshold at the 99.9th percentile of spot intensities observed in Δ*rtcB* cells. Fluorescent spots exceeding the false-positive threshold were classified as specific signals corresponding to *rtcB* mRNA molecules hybridised with complementary DNA probes. The probability distribution of the peak height and intensity of these fluorescent spots were extracted to determine the mRNA copy numbers. The intensity distribution of spots from a low-expressing control strain was fitted to a multi-Gaussian function, with the mean of the first Gaussian representing the intensity of a single mRNA molecule. The total fluorescence intensity of spots in each cell was divided by the intensity of a single mRNA molecule to calculate the number of mRNA molecules per cell. These data were then used to calculate the relative frequencies, mean and standard deviation of mRNA copy numbers across the population.

### Statistical analysis of smRNA-FISH data.

We used a one-sided *z*-test to confirm that mRNA copy numbers across *rtcB*-expressing cells were significantly higher under induced ($N = 75$) than under uninduced

conditions ($N$ = 39, $p$-value = 0.0003). To further estimate the proportion of *rtcBA*-induced cells, we employed a stochastic model of $\sigma^{54}$-controlled transcription[66] and accordingly fitted zero-inflated negative binomial distributions using the statsmodels package in Python.

**Determination of translation elongation speed in vivo.** The speed of translation elongation was measured in vivo via the production of a FusA-LacZα fusion protein encoded on plasmid pBBR-fusA-lacZα[11,67]. The strains were grown in LB supplemented with 10 µg/ml of gentamicin to select for the reporter plasmid and in presence or absence of 1.5 µg/ml of tetracycline as described above. FusA-LacZα expression was induced by 5 mM IPTG. Aliquots of 500 µl were taken every 20 seconds, translation was blocked using 10 µl of 34 mg/ml chloramphenicol, the samples were frozen in liquid nitrogen and stored at −80 °C. To assess FusA-LacZα production, 400 µl of each sample were incubated for 1 h at 37 °C, then treated for 2 h with 100 µl of 5x Z-Buffer (60 mM Na$_2$HPO$_4$, 40 mM Na$_2$HPO$_4$, 10 mM KCl, 1 mM MgSO$_4$, 50 mM β-mercaptoethanol, pH7) and 100 µl of 4mg/ml 4-methylumbelliferyl-D-galactopyranoside[11,67]. The reaction was stopped by adding 300 µl of 1 M Na$_2$CO$_3$ and the fluorescence intensity at 450 nm emission was measured after excitation at 365 nm. The square root of newly synthesised FusA-LacZα was plotted against the induction time. The translation time of FusA-LacZα was calculated as the $x$-intercept and corrected by the time it takes to initiate translation (measured as 10 s by Zhou et al.[67]). The translation elongation speed was obtained by dividing the length of FusA-LacZα (774 amino acids) by the corrected translation time. The Julia GLM.jl (v1.9.0) package was used to fit linear regression models to translation elongation speed[68].

**Sensitivity analysis.** For our parameter sensitivity analysis we studied parameter ranges that we found relevant from literature sources. The full set of parameters used in the model can be found in the SI. We varied ATP over the range 1000–5000 µM according to known concentrations of ATP in *E. coli*[69]; $\lambda$ was varied between 0.01–0.02 min$^{-1}$, corresponding to doubling times between approximately 30–70 min and so representative of intermediate to fast growth in *E. coli*[70].

**Preparation of cell samples for Raman microspectroscopy.** *E. coli* K12 strain BW25113 wildtype and deletion mutants from the KEIO collection of *rtcA* (cat no. OEC4987-213607960, Horizon Discovery), *rtcB* (cat no. OEC4987-213607484, Horizon Discovery), and *rtcR* (cat no. OEC4987-200828461, Horizon Discovery) were grown in LB and treated with and without 1.5 µg/ml tetracycline. For each condition, the same number of cells were centrifuged at 4 °C, 5000 × g for 2 minutes and mixed with 3.7% formaldehyde in ice-cold PBS buffer to metabolically fix the cells. The cell suspensions were shaken at 270 rpm at room temperature for 30 minutes. Fixed cells were collected by centrifugation at 400 × g for 8 minutes. After two washes with PBS followed by centrifugation at 600g for 3 minutes, cells were resuspended in distilled water at OD$_{600}$ ~0.05 for analysis. A volume of 2 µL of each cell sample was suspended on a stainless-steel microscope slide (Elliot Scientific) and air-dried prior to Raman microspectroscopy.

**Preparation of the reference library for Raman microspectroscopy.** A reference library was created to deconvolute the cellular Raman spectra using linear combination. The library comprised individual Raman spectra measured from the following biomolecules: total DNA, RNA and protein from *E. coli* BW25113 wildtype cells, commercially available aspartate, ATP, fructose-1,6-bisphosphate (FBP), glucose, glutamate, glutamine, glutathione (GSH), GTP, lignin (absent in bacteria hence serving as negative control), *E. coli* polar lipid extract, NAD+, UTP and valine. Total cellular DNA, RNA, and protein was extracted and purified by the AllPrep DNA/RNA/Protein Mini kit (cat no. 47054, Qiagen) following manufacturer's instructions. Total DNA and RNA were eluted with

distilled water. Total protein was eluted with 1% SDS in 10 mM HEPES and further purified by acetone precipitation to eliminate SDS and buffer. Typically, 100 µL protein in 1% SDS/10 mM HEPES was mixed with 400 µL acetone at 4 °C and kept at −20 °C overnight. The protein suspension was centrifuged at 13,000g for 10 minutes at 4 °C and washed again with acetone at 4 °C. After centrifugation, acetone was carefully removed by pipetting and volatilised at room temperature. The protein was resuspended in distilled water under constant vortexing. The remaining biomolecules were purchased in purified form. All reference biomolecules were dissolved in distilled water at 10 mg/ml and adjusted to pH 7-8. A volume of 2 µL of each sample was suspended on a stainless-steel microscope slide (Elliot Scientific) and air-dried prior to Raman microspectroscopy.

**Raman microspectroscopy.** Raman microspectroscopy was performed using an upright inVia™ confocal Raman microscope (Renishaw) with a 50x objective (0.8 numerical aperture), HeNe laser at 633 nm excitation wavelength and 200 seconds exposure time. The 1 µm laser spot was focused on individual *E. coli* cells via the point-selection mode of the mapping measurement in the WiRE 5.2 interface (Renishaw). Each spectrum was collected by accumulating the signal from 20 iterations of 10 second exposure to 100% laser power. For measurements of the purified biomolecules, the laser spot was focused on the perimeter of ring-like deposits from air-dried liquid drops. The Raman spectra of the purified biomolecules were the average of at least three independent measurements. The fingerprint region in the wavenumber range of 600–1800 cm$^{-1}$ was used for analysis of the Raman spectra.

**Background subtraction and data processing after Raman microspectroscopy.** Data processing and analysis was carried out in MATLAB 2022R. After removal of cosmic rays and smoothing by a Savitzky-Golay (SG) filter (window size = 11, polynomial order = 3), the fluorescent background was subtracted via multiple lines. Typically, the three lowest points at the start, middle, and end of the spectrum were identified and connected by two straight lines. The lowest point near the intersection where the lines cross the spectrum was identified and connected with other points to form new lines. This iteration was performed until no more intersections were identified.

The cellular spectra were decomposed by linear combination of reference spectra from purified biomolecules including DNA, RNA and protein extracted from the cells using the AllPrep DNA/RNA/Protein Mini kit (Qiagen) and fitted using the least squares method. The Raman intensity of each biomolecule $i$ was quantified by fitting their corresponding coefficients $p_i$ in the total cell spectrum via

$$\text{cell spectrum} = p_1 x_1 + p_2 x_2 + \ldots + p_i x_i, \qquad (9)$$

(Fig. S12). Here, $x_i$ are the normalised Raman spectra of each purified biomolecule, and the coefficients $p_i$ quantify the relative contributions of the normalised Raman spectra to the total cell spectrum. The standard error and confidence interval of coefficients were determined by computing the residuals and parameter covariance matrix. Kullback-Leibler divergence was used to quantify the difference between cellular and fitted Raman spectra.

**Statistical analysis of Raman microspectroscopy data.** We used a two-sided unpaired two-sample t-test to compare the means of RNA/protein ratios of each knockout strain to the WT. We further used a one-sided Mann Whitney U test to check if the distributions of *rtc* knockout samples displayed significantly lower RNA/protein ratios as well as RNA level than WT samples. Using one-sided Mann Whitney U, we also tested if the distribution of total protein was higher in knockout than in WT samples. We used the HypothesisTests.jl

(v0.11.3)[71] for all statistical testing. The KernelDensity.jl (v0.6.9) package[72] was used to fit kernel density estimates of RNA/protein histograms.

## Reporting summary

Further information on research design is available in the Nature Portfolio Reporting Summary linked to this article.

## Data availability

The data generated in this study are provided in the accompanying source data file. Source data are provided with this paper.

## Code availability

The code used to develop the model, perform the analyses and generate results in this study is made publicly available and has been archived in Zenodo under an MIT license. It is accessible via https://doi.org/10.5281/zenodo.17048324[57] or https://github.com/hhindley/rtc_model. The code for smoothing, background subtraction and deconvolution of the cellular Raman spectra is available via GitHub at: https://github.com/Engl-lab/Cellular-Raman.git.

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

## Acknowledgements

H.J.H. and A.Y.W. were supported by the Biotechnology and Biological Sciences Research Council, BBSRC (BB/Y004035/1) and C.E. by BBSRC (BB/W019698/1), Leverhulme Trust (RPG-2021-050), and Wellcome Trust (WT 213955/Z/18/Z). I.K.L. was supported by Leverhulme Trust (RPG-2019-092). Z.G. was supported by a PhD scholarship (award 202206950021) from the China Scholarship Council. We also thank Meriem El Karoui, Elena Pascual García, Diego Oyarzún and Peter Swain for insightful discussions and feedback.

## Author contributions

H.J.H. executed the research, A.Y.W. conceived and supervised the work, and H.J.H. and A.Y.W. wrote the manuscript. I.K.L., M.B. and C.E. contributed to the model development and to interpretation of results. C.E. conceived and supervised all experiments with help from AS for Raman spectroscopy. Z.G. conducted the Raman, SM the smRNA-FISH and MGG the LacZα-complementation experiments. All authors read and revised the manuscript.

## Competing interests

The authors declare no competing interests.
