## [Transparent Peer Review file · Nature Communications]

Heterogeneity in responses to ribosome-targeting antibiotics mediated by bacterial RNA repair

Corresponding Author: Dr Andrea Weisse

Version 0:

Reviewer comments:

Reviewer #1

(Remarks to the Author)
Hindley et al.

Mechanistic models of Rtc-regulated RNA repair suggests molecular targets to potentiate antibiotic effects.

The authors present a complex mathematical model to account for repair of ribosomal RNAs and tRNAs by the bacterial Rtc system, comprising the RNA ligase RtcB, the RNA cyclase RtcA, and the transcription activator RtcR that controls their expression.

Although the aims are ambitious, I find that the study is flawed in a many key respects, including the “facts” and assumptions that underpin the model and the unwarranted generalization to diverse bacterial taxa.

In the Introduction, the authors give great weight to the “crucial” role of RtcB in the repair of rRNA suggested by the study of MazF by Moll and colleagues (ref 12). Yet, they seem to ignore, or at least do not cite, multiple subsequent studies that effectively vitiate the Moll findings and their interpretation: e.g., Mets et al. 2017 RNA Biol 14, 124; Culviner and Laub 2018 Mol Cell 70, 868; and Mets et al. 2019 Biochimie 156, 79.

A central assumption of the model, as depicted in Fig. 1B, is that the RNA damage that incises the phosphodiester backbone leaves a 3'-PO₄,2'-OH end that must be cyclized by RtcA to form a 2',3'-cyclic phosphate end that serves as the substrate for end joining by the RtcB ligase. This is not a valid assumption. Indeed, the endonuclease ribotoxins that have been studied as agents of RNA incision acts via a transesterification mechanism that directly yields 2',3'-cyclic phosphate and 5'-OH ends (not a 3'-PO₄,2'-OH as assumed). Moreover, even if it were the case that a damaging ribonuclease did directly yield a 3'-PO₄,2'-OH end, there would be no need whatsoever for RtcA to bring about RNA repair by RtcB – because the RtcB mechanism entails the joining of 3'-PO₄ and 5'-OH ends via an RNA 3'pp5'G intermediate. In fact, RtcB has an intrinsic cyclic phosphodiesterase activity that hydrolyzes a 2',3'-cyclic phosphate to a 3'-PO₄,2'-OH that serves as the immediate substrate for ligation. The authors cite the relevant papers (refs 25-29) but don't assimilate the mechanistic points into their model. It is not clear why they “model the action of RtcB only on 2',3'-cyclic phosphate termini” (no evidence that this is so) or why they “consider RtcA as necessary to produce the cyclic substrate for RtcB” (ditto).

The model focuses on repair of RNA damage within otherwise intact ribosomes, which they classify as healthy (unbroken), damaged (3'-PO₄,2'-OH breaks), or tagged (2',3'-cyclic phosphate ends formed after the action of RtcA, which can be repaired by RtcB. For reasons outlined in the preceding paragraph, I do not see the justification for invoking the “tagged ribosome” state as the consequence of RtcA action on “damaged” ribosomes with 3'-PO₄,2'-OH breaks.

The authors are well justified in favoring a model (developed by Sandy Wolin's lab and others) whereby 2',3'-cyclic phosphate tRNA breaks are the signal that activates (derepresses) the RtcR transcription factor via interaction with its CARF domain.

They automatically extend this idea to breaks in rRNA and proceed to invoke cooperative binding of six ligands for full RtcR activation. This seems like a stretch. How would that work? Does RtcR need to bind simultaneously to six ribosomes, each of which has a surface accessible 2',3'-cyclic phosphate break? Or does RtcR bind to a ribosome with multiple 2',3'-cyclic

phosphate breaks?

The focus in this model on rRNA damage seems at odds with the weight of evidence that the RtcB product of the *E. coli* and *Salmonella* *rtcBA* operon is dedicated to tRNA repair. It would be important to cite such studies, e.g., on Rtc repair of damage caused by the *Salmonella* tRNA-Leu anticodon nuclease (Uppalapati et al. 2024 *Science* 384, 100).

A major omission is consideration of the fact that some bacteria encode more than one RtcB ligase. In particular, the work of Raven Huang's lab merits emphasis (PNAS 2022 119, e2202464119). They show that some strains of *E. coli* have a second RtcB ligase (RtcB2) encoded in an operon with PrfH. And they convincingly show that there is a mutually exclusive division of labor between the two RtcBs, whereby RtcB1 (the original RtcB) is dedicated to repair of tRNA breaks inflicted by colicins D and E5 whereas RtcB2 is dedicated to repair of 16S rRNA breaks (in the decoding center) inflicted by colicin E3 and CdiA.

Because RtcB2, the ribosome repair ligase, is not in an operon flanked by RtcR, one assumes that it is not subject to the same transcriptional controls as RtcB1. In that case, the whole premise here that the authors are modeling repair of RNA damage to the ribosome by the *rtcR*•*rtcBA* system may be invalid from the start.

Also, taking into account that Huang identifies 60 strains of *E. coli* that have a third RtcB paralog, plus the fact that *Myxococcus* has six RtcB paralogs, the model for control of bacterial RNA repair likely underestimates the scope and complexity of the problem.

(Remarks on code availability)

Reviewer #2

(Remarks to the Author)

The authors present and analyze a computational model of Rtc-regulated maintenance of long-lived RNAs, such as rRNA in ribosomes and tRNAs. Their findings indicate that the positive regulation of the RtcAB operon by RtcR results in a wide range of bistability in response to increasing RNA damage rates. Additionally, their computational analysis suggests that inhibiting RtcB and RtcR, but not RtcA, can abolish heterogeneity and potentiate the effects of antibiotics.

I would like to review this manuscript in 4 parts: methodological soundness, novelty, evidence, relevance.

Methodological soundness:

The authors present an ODE model that captures the dynamics of healthy, tagged, and damaged ribosomes, along with the dynamics of RtcA, RtcB, and R. Before delving into the specifics of how the terms and parameters are modeled, I would like to state that overall, the modeling effort is well thought out, and the analysis and figures are well made. The main point of the paper is that this system demonstrates robust bistability, which can be disrupted by inhibiting RtcB and R. The suggestions I will list, if implemented, would likely not violate the bistability of the system.

Specific comments:

- 1) The growth rate is modeled as the Greek letter lambda, which is taken as a constant. However, it is well known that the growth rate is almost linearly dependent on the number of healthy ribosomes.
- 2) The influx of healthy ribosomes, referred to as ν_{in} , is also treated as a constant. In the supplementary materials, it is related to κ_{in} , the influx of healthy RNAs. There are two modeling efforts in the manuscript: one for ribosomes and one for tRNAs. The term ν_{in} is used in both models. However, for the synthesis of ribosomes, healthy ribosomes should be considered, as they are composed of proteins and rRNA.

As I mentioned, these changes probably would not annihilate bistability, which brings me to one of the core discussions about this manuscript

Novelty:

The bistability in this case results from the positive regulation of the RtcAB operon by RtcR. It is a well-known fact that positive feedback regulation leads to bistability, similar to the lac operon (for whole-cell stochastic simulations, see Roberts, Elijah, et al. "Noise contributions in an inducible genetic switch: a whole-cell simulation study." *PLoS computational biology* 7.3 (2011): e1002010). Many other examples can be provided.

A different kind of bistability is mentioned in a paper already cited by the authors: Deris, J. Barrett, et al. "The innate growth bistability and fitness landscapes of antibiotic-resistant bacteria." *Science* 342.6162 (2013): 1237435. This paper argues the following:

"In particular, translation-inhibiting antibiotics have been shown to reduce the expression of both regulated and constitutively expressed genes because of growth-mediated global effects (16, 17). If one of these gene products provides some degree of antibiotic resistance, then growth inhibition can reduce expression of resistance; the diminished resistance can in turn allow the drug to further inhibit growth in a positive feedback loop (fig. S1), driving the cell into a stable nongrowing state after a transient slowdown in cell growth. Frequently, gene regulatory systems with positive feedback exhibit a switchlike behavior when, for example, intrinsic fluctuations in gene expression exceed some threshold (19, 20). This is often accompanied by bifurcation of a genetically homogeneous culture into two subpopulations with distinct

phenotypes, which is called bistability (19, 20). In the context of antibiotic resistance, this would be manifested as a “growth bistability,” that is, growing and nongrowing cells coexisting in a homogeneous environment”

First and foremost, the positive feedback regulation in the Rtc system (RtcR activating RtcAB, RtcA tagging ribosomes, tagged ribosomes activating RtcR) ensures bistability. Even without this positive feedback, incorporating growth effects into the model, such as a non-constant lambda, would still lead to bistability. This is because any protein produced by healthy ribosomes that confers resistance to antibiotic effects would exhibit bistability.

In the latter paper, they go the extra mile to test two systems: CAT expression (which degrades chloramphenicol) and the Ta1 strain (which constitutively exports tetracycline). Both were measured in single-cell experiments and demonstrated bistability, as highlighted by the statement: “These results were similar to those for Cat1 cells in Cm, supporting the hypothesis that growth bistability occurs generically, independent of the mode of drug resistance, as is predicted by growth-mediated feedback”.

Specific comment:

3) The two bistabilities that I mentioned are different: one results from the regulation, the latter results from the growth-dependent effects, contrasting these two and describing how each relates to Rtc system.

Nevertheless, the manuscript goes on to identify two of the three proteins of interest as potential targets to potentiate antibiotic effects.

Specific comment:

4) If it is known that the Rtc system is important for stress management or recovery from a stressed state and that it only comprises three proteins, targeting the entire Rtc system may be more effective than targeting just two members (RtcA and R). Since not all aspects of the Rtc system are fully understood, and for example that the Rtc system is known to affect mobility (as noted in a paper by one of the authors), it raises the question of why targeting the whole system might be considered inferior to targeting only RtcA and R. 3 out of 3 versus 2 out of 3

Evidence,

So, the existence of bistability is almost guaranteed, which raises concerns about the novelty of this finding. Nevertheless, it is important to determine if some of the observed bistabilities are specifically due to the Rtc system. Understanding the role of the Rtc system in general bistabilities would be particularly interesting and valuable.

However, this paper only hypothesizes bistability if the system is modeled in the specified way and with parameters from the literature. The authors state, “Our results suggest that the response may be underpinned by a form of hetero-resistance, and they suggest a granular characterization, at the level of single cells, will be necessary to test the hypothesis.” The only evidence presented in this paper is the expression levels of the Rtc system (Supplementary Figure 1), which are very low.

Specific comment:

5) I see how expression levels lower than 1 copy per cell indicate that some cells have 0 copies, but this does not necessarily indicate bistability. A gene that is very lowly expressed and does not show bistability could produce similar results.

The model is not used to emulate any experimental data (levels of Rtc members, etc).

Specific comment:

6) The model is not validated any experimental data? How is this model validated?

Since there not much experimental evidence, I have looked elsewhere. I have been troubled with the conflict with experiments in other papers and the presumed role of Rtc system.

Specific comment:

7) In my understanding of this manuscript, the wild-type bacteria with an Rtc system should show bistability. However, according to experiments in “Deris, J. Barrett, et al. ‘The innate growth bistability and fitness landscapes of antibiotic-resistant bacteria.’ *Science* 342.6162 (2013): 1237435,” the wild-type bacteria do not show bistability when treated with chloramphenicol (which is also indicated to induce the Rtc system, as reported in Engl, Christoph, et al. ‘Cellular and molecular phenotypes depending upon the RNA repair system RtcAB of *Escherichia coli*.’ *Nucleic Acids Research* 44.20 (2016): 9933-9941). How can we reconcile these two seemingly conflicting results?

Relevance,

According to one of the cited papers, “Manwar, Muhammad Ramzan, et al. ‘The bacterial RNA ligase RtcB accelerates the repair process of fragmented rRNA upon releasing the antibiotic stress.’ *Science China Life Sciences* 63 (2020): 251-258,” the authors state: “Therefore, rtcB is widely distributed but not highly prevalent across bacterial lineages, and such features suggest possible non-essential physiological functions which are perhaps advantageous under specific environmental conditions”. The prevalence that they give are very low.

Specific comment:

8) If the RtcB is not very prevalent, how can targeting it would be clinically relevant?

Overall,

In my understanding, after reading other papers by the authors (who are experts on Rtc), I am persuaded that bistability may exist for some bacteria in response to certain drugs, and it may be important to understand the role of the Rtc system in this

context. However, the evidence remains hypothetical. Although the manuscript is eloquently handled and represents a good modeling effort, it lacks substantial validation and evidence, remaining a theoretical exploration that highlights the potential importance of the Rtc system.

(Remarks on code availability)

Reviewer #3

(Remarks to the Author)

This is a very interesting paper, developing a mathematical model of the Rtc RNA repair system, which the authors (convincingly) argue could be used in the fight against antibiotic resistance. Their findings suggest several predictions to be tested experimentally, as well as proposed treatments against bacterial infections.

The paper is well-written, the maths are clear and the figures are pretty. However, I have several questions I'd like the authors to address:

1) What happens if RtcB can work directly on damaged ribosomes, as you mention in line 161, page 2? It seems that it wouldn't change much qualitatively, right?

2) In eqs. 1-3, shouldn't growth rate be a function of healthy ribosomes, according to Scott and Hwa's growth laws, or the corresponding author's own model published in PNAS in 2015? As healthy ribosomes go down, so does growth rate, and this introduces a nice feedback that could lead to a nice expression for growth rate as a function of the parameters (similarly to what Greulich et al did using these growth laws: <https://www.emboexpress.org/doi/pdf/10.15252/msb.20145949>).

In fact, if you take v_{in} , v_{rep} and v_{tag} as linear, and write $r_h = \lambda / K_t$ (Scott's first growth law) then you can find expressions for all three ribosome species as a function of λ and the repair parameters (which are a function of p_A and p_B and thus of growth rate as well, in equilibrium), and sum all three ribosome equations to obtain one polynomial equation for growth rate as a function of the parameters. You don't need to do this last part, but I'd like to see how the model changes if you include the feedback between ribosomes and growth rate. Or at least mention this as an extension of the model in the Discussion.

3) The main text should explain that v_0 in Figure 2B is a linear function of p_R^* . The authors explain this with words in the main text, and show the equations in the Supp, but adding the equation for v_0 in the main text after the explanation (in line 275) would clarify things.

4) The references to Cleland 1975 (ref 4 in Supp) has errors. Also ref 16 has too many caps!

5) Eq 18 in the Supp should end with r_h , not r_d

6) At the beginning of page S3, the definitions of K_a and K_b are reversed.

7) The organization of Supp Figures is confusing. I suggest getting rid of S5A, which is not very informative (protein levels are proportional to mRNA levels, and so there is no need to show this, I think). S5B (which is not mentioned in the main text) seems to be identical to 3B, and should be removed. Then the text naturally cites S1 first. Supp Figure S5C should become S2, and the actual S2 would then become S3. Supp Fig S5D should become a separate figure, S4. Current figures S3 and S4 would then become S5 and S6.

8) In L368, "This finding is backed-up by gene expression data (Figure S1), where average molecule numbers of the rtc mRNAs are well below one in WT and inducing conditions, suggesting that in a population of cells, not all cells are expressing rtc. In inducing conditions, upon exposure to ribosome-targeting antibiotics, this could suggest a heterogeneous response."

Where is this data from? Is this from your lab? I don't see any description of the Methods used.

9) In Fig 3C, what are the values of λ and ω_{ba} used in the first plot, the values of ATP and ω_{ba} in the second plot, and the values of ATP and λ used in the third plot? Also, this figure looks too crowded, I'd suggest adding more padding between subplots. Alternatively, make it a double-column figure.

10) In Fig S5C, the grey lines should be dashed in the unstable branch.

11) I'm not sure I understand Fig S2. You claim that 99% of parameter combinations yield bistable responses, but for all parameters, the area of the violin plots yielding monostable responses is the same as the area yielding bistable responses? Is this a mistake?

12) In Fig 4A, I find it strange that the values leading to the OFF state are so few. I would have expected all initial conditions below the unstable branch should lead to the lower stable point. Why doesn't this happen? Can you do a phase-plane

analysis to show this?

13) In L547: "Finally, we assumed that the rate of influx of healthy ribosomes is proportional to the rate of translational activity exerted by ribosomes themselves."

What is the rationale behind this?

(Remarks on code availability)

I haven't downloaded the code, but it seems pretty straightforward to download and use. The README file gives enough instruction to use the code.

Version 1:

Reviewer comments:

Reviewer #1

(Remarks to the Author)

REVISED ms. Hindley et al.

The authors refer in the Introduction to "Rtc" as a "system" comprising both RtcB (ligase) and RtcA (cyclase). They then say that "Rtc" functions in metazoan/archaeal tRNA splicing. But this is not strictly the case, insofar as the RtcB ligase is clearly a tRNA splicing enzyme in those taxa but (as far as this reviewer is aware) there is no evidence that RtcA plays any role in tRNA splicing.

I had previously objected to central assumption of the model, as depicted in Fig. 1B, that the RNA damage that incises the phosphodiester backbone leaves a 3'-PO₄,2'-OH end that must be cyclized by RtcA to form a 2',3'-cyclic phosphate end that serves as the substrate for end joining by the RtcB ligase. This is still not a valid assumption. RtcB is just as active in sealing a 3-PO₄ end as a cyclic PO₄ ends. In their rebuttal, the authors cite published findings that the rate of single-turnover 3-PO₄ cyclization by RtcA is much faster than the rate of ligation by RtcB as a validation of their assumption that any 3'-PO₄ end will be cyclized prior to encounter with RtcB. But this is off point. Single-turnover kinetics are performed in enzyme excess. In the cellular milieu, the partition of 3'-PO₄ ends between cyclization and ligation will depend on the concentrations of RtcA and RtcB (not known?) and the relative affinities of RtcA and RtcB for their respective 3' end substrates in the context of damaged tRNAs and rRNAs of concern in this paper (also unknown?).

Therefore, I consider the statement on lines 177-180 ("When the two are co-expressed, as in the antibiotic response we consider, RtcA thus likely removes reactive 3'-phosphate ends before RtcB can, and so in the model we only consider the action of RtcB on 2',3'-cyclic phosphate termini.") as unjustified.

The heart of the matter is whether the cyclase activity of RtcA is necessary for RNA repair by RtcB, as the authors predict and assume in Fig. 1B. The mere fact that the *rtcB* and *rtcA* genes are coexpressed and co-induced does not establish this point. The authors state in rebuttal that " Δ *rtcA* and Δ *rtcB* strains display impaired growth compared to wildtype *E. coli*, and complementation with RtcA or RtcB, respectively, recovers wildtype growth (Engl, Schaefer, Kotta-Loizou & Buck, 2016, *Nucleic Acids Res.*, 44), implying that both are required for full growth rescue in the conditions we consider." However, the prior NAR paper does not establish the point regarding RtcA. For example, the NAR paper did not establish whether the phenotypes attributed to Δ *rtcB* and Δ *rtcA* reflect lack of the proteins versus lack of their RNA repair catalytic activities. RtcA and RtcB both act via a mechanism of covalent nucleotidyl transfer to an active site histidine nucleophile. The active sites have been defined, so it would be straightforward to compare complementation of Δ *rtcB* and Δ *rtcA* phenotypes by wild-type and His-to-Ala mutant alleles of *rtcB* and *rtcA*. The NAR paper demonstrated fragmentation of rRNA in Δ *rtcB* cells (presumably owing to a defect in rRNA repair) that was rectified by expression of RtcB. I will grant that it is overwhelmingly likely that an active site mutant of RtcB will not complement this phenotype. Yet no such analysis was presented for Δ *rtcA* cells so we are in the dark as to whether lack of RtcA elicits rRNA fragmentation (an rRNA repair defect) and whether this is because of a lack of cyclase activity. (In this vein, it is worth noting that expression of *E. coli* RtcB in yeast in lieu of yeast Trl1 complements tRNA splicing and Hac1 splicing in the UPR, in the absence of any RtcA.) Thus, I do not agree with the authors' assertion that "our assumption that RtcA has a role in the repair response seems well justified."

The authors acknowledge (lines 167-170) that RtcA can also convert 2'-PO₄ ends into a cyclic-PO₄ that can be repaired by RtcB, but they dismiss this reaction from consideration in their model because it is much slower than 3'-PO cyclization. This seems arbitrary and perhaps unwise. As noted above, there is no a priori reason for RtcB to "care" whether it is ligating a 3'-PO₄ or cyclic-PO₄ end. But if RNA damage produces a 2'-PO₄ end, either directly or when the *E. coli* CPDase ThpR converts a cyclic-PO₄ end into a 2'-PO₄ end (PMC4201822), there is no way for this 2'-PO₄ end to be sealed by RtcB unless RtcA cyclizes it.

This highlights a central knowledge gap in that we do not know what types of RNA ends are generated during the damage in the antibiotic response that the authors invoke. In many well-studied instances (ribotoxin endonucleases that incise RNA via transesterification), the immediate product is a cyclic-PO₄.

The authors have tempered their assumptions that rRNA repair is the prime function of RtcAB and now extend the activity to spectrum to broken tRNAs (for which there is more persuasive evidence). Yet I still have difficulty accepting their

classification of ribosomes (or tRNAs) as healthy (unbroken), damaged (3'-PO₄,2'-OH breaks), or tagged (2',3'-cyclic phosphate ends formed after the action of RtcA), whereby the tagged states can be repaired by RtcB. I do not see the justification for invoking the "tagged ribosome" (or tagged RNA) state as the consequence of RtcA action on "damaged" ribosomes (or tRNAs) with 3'-PO₄,2'-OH breaks, as opposed to such ends being "damaged" themselves.

The authors include new experiments that image single cells for *rtcB* expression by FISH before/after exposure to tetracycline and indicate that there is cell-to-cell heterogeneity whereby only some cells are induced. They provide evidence that translation speed is reduced in Δ *rtcB* cells in the absence of antibiotic. What really needs to be done here is to perform the same translation speed experiment for Δ *rtcA* cells (while also doing the rRNA damage assay on Δ *rtcA* cells as in the NAR paper; vide supra). In other words, test the assumption that RtcA (and its cyclase activity) is important as posited.

A concern here is that it is not obvious how to account for the global slowing in translation in Δ *rtcB* cells versus wild-type given that the vast majority (~90%) of wild-type cells have no *rtcB* mRNA (detectable by FISH) to begin with. Could it be the case that the FISH assay underestimates the mRNA level in individual cells scored as having 0 transcripts.

I had raised a question about the allosteric model for activation of the RtcR hexamer. The authors state (lines 281-284) that this requires "the cooperative binding of six ligands [RNA with 2',3'-cyclic termini] for full activation." Why can't allostery be achieved by binding of one or two ligands? The statement in the rebuttal ("the reviewer is correct that we currently do not know how many ligands are required for full activation, and thus we cannot categorically state that six ligands are used by one RtcR hexamer") seems to be at odds with the categorical statement/assumption in the Results.

(Remarks on code availability)

Reviewer #2

(Remarks to the Author)

Summary

The final version of the paper argues that a model developed for the Rtc system supports robust bistability (heteroresistance). This conclusion is reinforced by new FISH experiments showing that only a small fraction of bacteria express Rtc; upon antibiotic exposure, however, this fraction grows. Moreover, Δ *RtcB* cells exhibit slower translation rates both with and without antibiotics, with a "slightly more pronounced effect in the presence of antibiotics," as argued by the authors. The authors further suggest that inhibiting RtcB and RtcR might be more effective in blocking this effect than targeting RtcA.

Overall, there is clear progress from the initial submission to the revised manuscript. Credit to the authors that the new experiments and their results are important on their own. However, I think there are several major issues.

First of all, throughout the manuscript, the model is used for its qualitative insights: its demonstration of bistability supports the contention that bacterial cells can exhibit heterogeneous responses.

An alternative approach could have involved systematically measuring certain rate constants or other things, comparing them against predictions, and thereby validating the model more rigorously. While such a quantitative approach may not be strictly necessary, it underscores the qualitative (rather than quantitative) approach of the manuscript.

Since the model is not validated quantitatively, it should be validated qualitatively. These considerations raise two important questions:

1. Do the authors fully stand by the claim of bistability as predicted by their model?
2. Do they assert that blocking RtcR and RtcB indeed leads to significant potentiation of antibiotic treatment? and Rtc A does not.

I consider that authors ought to satisfactorily answer these questions upon the qualitative treatment of the model. I will elaborate while going over the points that are answered by the reviewers.

Major Concerns and Authors' Responses

1. Assumptions on Growth Rate and Ribosome Influx
Addressed satisfactorily.

2. Cooperativity and Positive Feedback in RtcR Activation
The claims are toned accordingly.

3. Experimental Validation and Evidence for Heterogeneity

This ties into the first major question mentioned in the summary. Does the data strongly support bistability? Low expression levels can lead to most cells lacking mRNA copies and some having a few copies. It is not clear to me as it is. Can authors claim bistability from their data definitively? Or further experimentation is needed.

4. Clinical Relevance and Targeting the Rtc System

As mentioned in the manuscript, there may be many complementary mechanisms and even Rtc proteins that can replace the function of the studied Rtc system, but it is argued that this system is the dominant one. This is a subjective assessment. It is not clear that blocking RtcB or R is enough to block tolerance of the cell under antibiotic exposure. This is indirectly shown by the fact that the $\Delta RtcB$ cell line has a slower translation rate. How can we be sure that it is enough? Can this be contrasted with anything that definitively causes tolerance, and its inhibition is clinically relevant?

5. Blocking RtcB and R is better than RtcA

Now the authors have shown that blocking RtcB does indeed lead to a lower translation rate (we still don't know if that is comparable to other literature-relevant cases). Why not try $\Delta RtcR$ and $\Delta RtcA$ and see if the response to $\Delta RtcA$ is different compared to $\Delta RtcB$ and $\Delta RtcR$?

Minor comments: There were some typos, repeated words; please proofread.

(Remarks on code availability)

Reviewer #3

(Remarks to the Author)

Thank you for your detailed response. The authors have (more than adequately!) answered my queries. I recommend publication as is.

(Remarks on code availability)

Version 2:

Reviewer comments:

Reviewer #2

(Remarks to the Author)

The authors have adequately addressed my concerns and queries.

(Remarks on code availability)

Reviewer #3

(Remarks to the Author)

Nothing new to address on my part.

(Remarks on code availability)

We thank the anonymous reviewers for the time they took to carefully inspect our manuscript and give us their considerate feedback. Their comments and suggestions have substantially improved our revised manuscript, which now contains several new model analyses and, crucially, novel data from new experiments that validate the central conclusions of our model predictions.

The extent and importance of these improvements are reflected in changes to all sections including the SI (highlighted in red), a whole new Results section including a new main figure (Fig. 4), 5 new supplementary figures, a new title (“*A model of Rtc-regulated RNA repair reveals heterogeneous responses to ribosome-targeting antibiotics and suggests molecular targets to potentiate their effects*”) and two additional authors.

Please find a detailed point-by-point response below. We hope this response will demonstrate to the reviewers that all their concerns have been addressed to full satisfaction and that our revised manuscript makes a valuable contribution worthy of publication in *Nature Communications*.

Response to comments by Reviewer #1:

Comment 1.1: The authors present a complex mathematical model to account for repair of ribosomal RNAs and tRNAs by the bacterial Rtc system, comprising the RNA ligase RtcB, the RNA cyclase RtcA, and the transcription activator RtcR that controls their expression.

Although the aims are ambitious, I find that the study is flawed in a many key respects, including the “facts” and assumptions that underpin the model and the unwarranted generalization to diverse bacterial taxa.

We acknowledge that our final claim regarding suggested routes to enhance treatments via inhibition of Rtc may come across as an ‘unwarranted generalization to diverse taxa’. In the revised manuscript, we ensured to limit our conclusions to *E. coli* and its closest relatives that carry the *rtcBA* operon, which include several nosocomial *Enterobacteriaceae* such as *Salmonella*, *Klebsiella*, and *Shigella* species. We addressed this concern with more moderate statements in the Introduction, Results and Discussion, as well as throughout, specifying the ‘*E. Coli rtcBA*’ operon instead of ‘Rtc system’ more generally.

We address other aspects of this comment in our responses to the specific concerns below.

Comment 1.2: In the Introduction, the authors give great weight to the “crucial” role of RtcB in the repair of rRNA suggested by the study of MazF by Moll and colleagues (ref 12). Yet, they seem to ignore, or at least do not cite, multiple subsequent studies that effectively vitiate the Moll findings and their interpretation: e.g., Mets et al. 2017 RNA Biol 14, 124; Culviner and Laub 2018 Mol Cell 70, 868; and Mets et al. 2019 Biochimie 156, 79.

We apologise that our presentational choices have led to confusion regarding the generality of our assumptions. We would like to emphasise that our model analysis assumes damage to RNA involved in translation in general, which may affect rRNA or tRNA. To this end, we presented two separate analyses of the two scenarios (rRNA , tRNA damage) in our original submission. We

appreciate how our decision to relegate the analysis of tRNA-repair to the SI may have given the impression that we favour assumptions of rRNA-repair over that of tRNA-repair, which was not our intention.

In our revised manuscript, we overhauled the model presentation (see first Results section). We now present a unified model (see also updated Fig. 2) of damage and Rtc-mediated repair of translational RNA, where we highlight how damage to either rRNA or tRNA affect specific model components differentially. Importantly, both scenarios yield the same conclusions.

At this stage, we prefer taking a cautious view and not exclude the possibility of damage and repair of rRNA. It is true that the cited works (Mets et al. 2017 RNA Biol 14, 124; Culviner and Laub 2018 Mol Cell 70, 868; and Mets et al. 2019 Biochimie 156, 79) vitiated select findings from one of the references (Temmel et al. 2017 NAR 45, 8) we cited to motivate assumptions on the role of RtcB in maintaining ribosome health: they found that MazF can cleave rRNA at various sites, thus proving a conclusion by Temmel et al. wrong that cleavage is specific to the anti-Shine-Dalgarno sequence and creates a specialized pool of ‘stress-ribosomes’. Our model, however, neither assumes specificity of damage sites in rRNA nor necessarily an involvement of MazF in the antibiotic responses we consider, but simply there is damage to rRNA.

Crucially, Temmel et al. also showed that RtcB acts to restore full functionality in cleaved rRNA, and this conclusion remains unchallenged by the later works. Along with Temmel et al., we also cited additional studies, and our revised manuscript presents new data (see response to reviewer #2), all of which evidence a role of RtcB in improving translational efficiency and a capacity to repair rRNA. However, to avoid confusion, in the revised manuscript we improved the way we cite Temmel et al. stating that “*some evidence further suggests that RtcB improves ribosome health through the re-ligation of fragmented rRNA as well as accumulation of ribosomal damage in the absence of RtcB*”.

Comment 1.3: A central assumption of the model, as depicted in Fig. 1B, is that the RNA damage that incises the phosphodiester backbone leaves a 3'-PO₄,2'-OH end that must be cyclized by RtcA to form a 2',3'-cyclic phosphate end that serves as the substrate for end joining by the RtcB ligase. This is not a valid assumption. Indeed, the endonuclease ribotoxins that have been studied as agents of RNA incision acts via a transesterification mechanism that directly yields 2',3'-cyclic phosphate and 5'-OH ends (not a 3'-PO₄,2'-OH as assumed). Moreover, even if it were the case that a damaging ribonuclease did directly yield a 3'-PO₄,2'-OH end, there would be no need whatsoever for RtcA to bring about RNA repair by RtcB – because the RtcB mechanism entails the joining of 3'-PO₄ and 5'-OH ends via an RNA3'pp5'G intermediate. In fact, RtcB has an intrinsic cyclic phosphodiesterase activity that hydrolyzes a 2',3'-cyclic phosphate to a 3'-PO₄,2'-OH that serves as the immediate substrate for ligation. The authors cite the relevant papers (refs 25-29) but don't assimilate the mechanistic points into their model. It is not clear why they “model the action of RtcB only on 2',3'-cyclic phosphate termini” (no evidence that this is so) or why they “consider RtcA as necessary to produce the cyclic substrate for RtcB” (ditto).

The model focuses on repair of RNA damage within otherwise intact ribosomes, which they classify as healthy (unbroken), damaged (3'-PO₄,2'-OH breaks), or tagged (2',3'-cyclic phosphate ends formed after the action of RtcA, which can be repaired by RtcB. For reasons outlined in the preceding paragraph, I do not see the justification for invoking the “tagged ribosome” state as the consequence of RtcA action on “damaged” ribosomes with 3'-PO₄,2'-OH breaks.

We thank the reviewer for this valuable comment. Since the route of RNA damage by antibiotic exposure is yet unknown, we indeed cannot exclude that damage may directly yield 2',3'-cyclic phosphate ends, in which case RtcA would not be required for repair via RtcB.

We now included an analysis of direct repair via RtcB in the revised manuscript. The analysis demonstrates a steady-state response that is equivalent to the one with damage yielding 3'-phosphate ends and involvement of RtcA, i.e. bistability in *rtc* expression and translational capacity with largely the same steady-state values except for minor deviations in a small range of damage rates. There are also minor differences between the two scenarios in their transient behaviour, as steady states are approached more rapidly when damage directly yields the ligand that induces *rtc* expression. This new analysis – see Results (Rtc-regulated RNA repair), SI §S1.5 and Fig. S1 – enhances the generality of our results and thus the applicability of our conclusions overall.

Regarding direct ligation of 3'-phosphate ends by RtcB, it has indeed been shown *in vitro* that RtcB has this capacity in addition to its widely recognised capacity to repair 2',3'-cyclic phosphate ends (Tanaka et al. 2011 JBC 286, 50). But how relevant direct repair of 3'-phosphate ends by RtcB is in physiological conditions when *rtcBA* are co-expressed is unclear: with RtcA processing 3'-phosphate ends almost 3 orders of magnitude faster (Chakravarty, Smith & Shuman, 2011, *Proc. Natl. Acad. Sci. U.S.A.*, 108) than RtcB (Maughan & Shuman 2016 J Bacteriol 198), quite plausibly, RtcA rapidly removes reactive 3'-phosphate ends before RtcB can. When RtcA and RtcB are co-expressed, as is the case in the antibiotic responses we consider, a likely route of repair therefore is the concerted 'healing' by RtcA and 'sealing' by RtcB as considered in our original submission (cf. Tanaka et al. 2011 JBC 286, 50).

Moreover, Δ *rtcA* and Δ *rtcB* strains display impaired growth compared to wildtype *E. coli*, and complementation with RtcA or RtcB, respectively, recovers wildtype growth (Engl, Schaefer, Kotta-Loizou & Buck, 2016, *Nucleic Acids Res.*, 44), implying that both are required for full growth rescue in the conditions we consider. It was further shown that RtcA is required for activation of RtcR and thus induction of the *E. coli rtcBA* operon (Kotta-Loizou et al., 2022, *iScience*, 25). Activation of RtcR requires the presence of a ligand, and the product of RtcA-mediated healing are 2',3'-cyclic phosphate ends, which we denoted as 'tagged' RNA and which are known to activate RtcR (Hughes et al., 2020, *Cell Rep.*, 33). We are not aware of evidence that suggests activation of RtcR by 3'-phosphate ends.

In view of the above and given that RtcBA are co-expressed from the same operon in *E. coli*, our assumption that RtcA has a role in the repair response seems well justified, however, our new analysis of direct repair via RtcB, when damage yields 2',3'-cyclic phosphate ends, also demonstrates that this assumption is not required for the conclusions we draw.

We acknowledge that the current body of knowledge around RNA repair in prokaryotes is incomplete. This is why we believe this first mathematical model makes a valuable contribution to the field by providing a testbed for competing hypotheses and a tool base to generate new hypotheses which can be explored experimentally. Our revised manuscript now includes a more elaborate discussion of uncertainties surrounding model assumptions and the role of mathematical models in the quest for resolving those (see first Results section and Discussion).

Comment 1.4: The authors are well justified in favoring a model (developed by Sandy Wolin's lab and others) whereby 2',3'-cyclic phosphate tRNA breaks are the signal that activates (derepresses) the RtcR transcription factor via interaction with its CARF domain. They automatically extend this idea to breaks in rRNA and proceed invoke cooperative binding of

six ligands for full RtcR activation. This seems like a stretch. How would that work? Does RtcR need to bind simultaneously to six ribosomes, each of which has a surface accessible 2',3'-cyclic phosphate break? Or does RtcR bind to a ribosome with multiple 2',3'-cyclic phosphate breaks?

We assume that up to six ligands can act to form the most active form of RtcR, based on evidence suggesting that RtcR acts as a hexamer (Jovanovic, Dworkin & Model, 1997, *J. Bacteriol.*, 179, Kotta-Loizou et al., 2022, *iScience*, 25). But the reviewer is correct that we currently do not know how many ligands are required for full activation, and thus we cannot categorically state that six ligands are used by one RtcR hexamer.

Following this comment, we analysed the impact the assumption has on model predictions. To this end, we analysed the sensitivity of steady-state predictions with respect to changes in the cooperativity coefficient and found that our conclusions are insensitive to the assumption (see sensitivity analysis in Results, Figs. S2D & S3D, and Discussion): lower cooperativity decreases steady-state expression levels of RtcBA and alters the steepness of induction in response to increasing damage, however, even in the most extreme case of no cooperativity, bistability is preserved.

We thank the reviewer for this important observation, which improved the manuscript and helped us gain more confidence regarding the robustness of our predictions.

Comment 1.5: The focus in this model on rRNA damage seems at odds with the weight of evidence that the RtcB product of the E. coli and Salmonella rtcBA operon is dedicated to tRNA repair. It would be important to cite such studies, e.g., on Rtc repair of damage caused by the Salmonella tRNA-Leu anticodon nuclease (Uppalapati et al. 2024 Science 384, 100).

We addressed this concern in our response to Comment 1.2, where we highlighted that we assume damage to RNA involved in translation more generally, with damage assumed to affect rRNA or tRNA, both of which yield the same conclusions. We thank the reviewer for the additional reference, which we have included in the revised submission (Uppalapati et al., 2024 *Science* 384, 100).

Comment 1.6: A major omission is consideration of the fact that some bacteria encode more than one RtcB ligase. In particular, the work of Raven Huang's lab merits emphasis (PNAS 2022 119, e2202464119). They show that some strains of E. coli have a second RtcB ligase (RtcB2) encoded in an operon with PrfH. And they convincingly show that there is a mutually exclusive division of labor between the two RtcBs, whereby RtcB1 (the original RtcB) is dedicated to repair of tRNA breaks inflicted by colicins D and E5 whereas RtcB2 is dedicated to repair of 16S rRNA breaks (in the decoding center) inflicted by colicin E3 and CdiA.

Tian et al. (PNAS 2022 119, e2202464119) demonstrated division of labour between the homologs, where RtcB2 repairs damage in ribosomes resulting from specific cleavage within the decoding centre in the small ribosomal subunit by two families of ribosome-specific ribotoxins, while RtcB1 that forms part of the RtcR-regulated *rtcBA* operon we consider efficiently repairs damage in tRNA. The work showed a clear difference in the targets of the two RtcB homologs for the specific damage they induced, which however does not rule out that RtcB1 can repair rRNA in other scenarios.

The specificities of different RtcB homologs are not fully understood. However, RtcB2 has been shown to act on disassembled ribosomes, while RtcB1 has been shown to interact with

assembled ribosomes (Manwar et al., 2020, *Sci. China Life Sci.*, 63). Given that RtcB1 associates with ribosomes and that knockout strains display impaired ribosome function (Temmel et al., 2017, *Nucleic Acids Res.*, 45, Manwar et al., 2020, *Sci. China Life Sci.*, 63), the results by Tian et al. do not provide sufficient evidence to conclude that repair via RtcB1 is confined to tRNA and thus that repair of rRNA is irrelevant in the scenario we consider.

*Comment 1.7: Because RtcB2, the ribosome repair ligase, is not in an operon flanked by RtcR, one assumes that it is not subject to the same transcriptional controls as RtcB1. In that case, the whole premise here that the authors are modeling repair of RNA damage to the ribosome by the *rtcR•rtcBA* system may be invalid from the start.*

Also, taking into account that Huang identifies 60 strains of E. coli that have a third RtcB paralog, plus the fact that Myxococcus has six RtcB paralogs, the model for control of bacterial RNA repair likely underestimates the scope and complexity of the problem.

We are currently not aware of evidence that suggest alternative RtcB homologs are induced in the antibiotic responses we consider, unlike RtcB1 as part of the *rtcR•rtcBA* system. It is possible that there is redundancy across homologs under some circumstances, which is a valid concern when aiming to potentiate antibiotic effects via inhibition of RtcB1. We now mention this possibility in the Discussion stating that “some bacteria, including several *E. coli* strains encode more than one RtcB ligase. While some of the homologs have been shown to repair complementary RNA substrates, we cannot exclude that there may be redundancy between them, which may under certain stresses limit the efficacy of inhibition.”

However, we also present new data (Fig. 4D & E), which demonstrate that lack of RtcB1 negatively impacts translation efficiency and that the effect is enhanced upon exposure to the inducing antibiotic tetracycline. The data therefore suggest a substantial, if not dominant, contribution of RtcB1 to growth rescue following distress with ribosome-targeting antibiotics.

Responses to comments by Reviewer #2:

Comment 2.1: Methodological soundness:

The authors present an ODE model that captures the dynamics of healthy, tagged, and damaged ribosomes, along with the dynamics of RtcA, RtcB, and R. Before delving into the specifics of how the terms and parameters are modeled, I would like to state that overall, the modeling effort is well thought out, and the analysis and figures are well made. The main point of the paper is that this system demonstrates robust bistability, which can be disrupted by inhibiting RtcB and R. The suggestions I will list, if implemented, would likely not violate the bistability of the system.

We thank the reviewer for highlighting the methodological soundness of our submission.

Comment 2.2: Specific comments:

1) The growth rate is modeled as the Greek letter lambda, which is taken as a constant. However, it is well known that the growth rate is almost linearly dependent on the number of healthy ribosomes.

We thank the reviewer for this valuable comment. A link between growth rate and ribosome abundance is indeed well established through a whole body of work on ribosomal growth laws spearheaded by Hwa and colleagues (Scott et al. 2010 *Science* 330-6007). In our model, we consider conditions of translational inhibition that are covered by the second growth law, where

growth rate and ribosome abundance exhibit a negative correlation. We now provide an extended analysis to the model where the growth rate is dynamically linked to the healthy ribosomes through the following equation: $\lambda_c * (r_{hmax} - r_h)$. We have addressed this alongside the influx of ribosomes from comment 2.3, please see a continuation of this response in the paragraph below.

Comment 2.3: The influx of healthy ribosomes, referred to as vin , is also treated as a constant. In the supplementary materials, it is related to kin , the influx of healthy RNAs. There are two modeling efforts in the manuscript: one for ribosomes and one for tRNAs. The term vin is used in both models. However, for the synthesis of ribosomes, healthy ribosomes should be considered, as they are composed of proteins and rRNA.

In addition to linking growth rate with dynamic ribosome levels (see previous comment), we also consider a dynamic influx of ribosomes in the extended analysis. Since ribosomes catalyse their own synthesis, we assume that influx rate is positively correlated with dynamic ribosome levels, where we include a saturating term to avoid exponentially growing influx: $(r_h * k_{in}^{max}) / (1 + r_h)$. Combining these two additional relationships (ribosomes with growth rate and influx) yields a model extension that we present in §S1.4. Steady-state and sensitivity analysis of this model, predicts a comparable bistable response, seen in Figure S7, to the one of the model with constant dilution and influx as presented in the main. In addition to the new supplementary section, we further address this comment in the Results section (see sensitivity analysis) and in the Discussion.

We thank the reviewer for these valuable comments and believe addressing them has solidified our conclusions regarding bistability in the *rtc* model.

Comment 2.4: Novelty:

The bistability in this case results from the positive regulation of the RtcAB operon by RtcR. It is a well-known fact that positive feedback regulation leads to bistability, similar to the lac operon (for whole-cell stochastic simulations, see Roberts, Elijah, et al. "Noise contributions in an inducible genetic switch: a whole-cell simulation study." PLoS computational biology 7.3 (2011): e1002010). Many other examples can be provided.

A different kind of bistability is mentioned in a paper already cited by the authors: Deris, J. Barrett, et al. "The innate growth bistability and fitness landscapes of antibiotic-resistant bacteria." Science 342.6162 (2013): 1237435. This paper argues the following:

"In particular, translation-inhibiting antibiotics have been shown to reduce the expression of both regulated and constitutively expressed genes because of growth-mediated global effects (16, 17). If one of these gene products provides some degree of antibiotic resistance, then growth inhibition can reduce expression of resistance; the diminished resistance can in turn allow the drug to further inhibit growth in a positive feedback loop (fig. S1), driving the cell into a stable nongrowing state after a transient slowdown in cell growth. Frequently, gene regulatory systems with positive feedback exhibit a switchlike behavior when, for example, intrinsic fluctuations in gene expression exceed some threshold (19, 20). This is often accompanied by bifurcation of a genetically homogeneous culture into two subpopulations with distinct phenotypes, which is called bistability (19, 20). In the context of antibiotic resistance, this would be manifested as a "growth bistability," that is, growing and nongrowing cells coexisting in a homogeneous environment"

First and foremost, the positive feedback regulation in the Rtc system (RtcR activating RtcAB,

RtcA tagging ribosomes, tagged ribosomes activating RtcR) ensures bistability. Even without this positive feedback, incorporating growth effects into the model, such as a non-constant lambda, would still lead to bistability. This is because any protein produced by healthy ribosomes that confers resistance to antibiotic effects would exhibit bistability.

In the latter paper, they go the extra mile to test two systems: CAT expression (which degrades chloramphenicol) and the Ta1 strain (which constitutively exports tetracycline). Both were measured in single-cell experiments and demonstrated bistability, as highlighted by the statement: “These results were similar to those for Cat1 cells in Cm, supporting the hypothesis that growth bistability occurs generically, independent of the mode of drug resistance, as is predicted by growth-mediated feedback”.

We did not intend to claim novelty on bistability arising from positive feedback. Our original submission acknowledged the association between positive feedback and bistability and cited relevant work (Ferrell, 2002, *Curr. Opin. Cell Biol.*, 14). We now highlight the association of bistability and positive feedback more clearly. The passage now reads *“hysteresis is a key feature of bistability and typically arises as a result of positive feedback, especially in combination with ultrasensitivity, both of which are present in the Rtc system.”*

The source of the misunderstanding is that positive feedback is necessary but not sufficient for bistability (Cinquin & Demongeot, 2002, *J Theor Biol* 216:229–241, Ferrell, 2002, *Curr. Opin. Cell Biol.*, 14). Outcomes of bistability generally depend on parameter choices, as also reflected in our sensitivity analysis (cf. Fig. S5 & S6) – *“positive feedback does not guarantee bistability”* (Angeli et al. 2004 *PNAS* 101 (7)).

Dynamically, growth-dependent feedback on a resistance-conferring factor presents another form of positive feedback (Barrett Deris et al. 2013 *Science* 342.6162). Given the above, its association with bistability is also more nuanced, as reflected in the passages quoted by the reviewer.

We are not aware of previous works that suggested bistability of Rtc-mediated responses – we ensured that any claims of novelty are confined to bistability of the Rtc system only.

Comment 2.5:

Specific comment:

3) The two bistabilities that I mentioned are different: one results from the regulation, the latter results from the growth-dependent effects, contrasting these two and describing how each relates to Rtc system.

Our new supplemental analysis with dynamically linked dilution and ribosome influx rates represents a situation with growth-dependent feedback (see response to Comments 2.2 & 2.3). We show this feedback also gives rise to bistability (Figure S7). Together with our main analysis, this suggests that the system displays comparable responses with and without growth-dependent feedback, and so positive feedback through the regulation of Rtc appears sufficient to produce the bistable response. We included a comment on contributions of the different feedbacks in the SI §S1.4.

Our additional analysis on sensitivity of bistable predictions with respect to the parameter that specifies the order of cooperativity of RtcR-activation (see response to Comment 1.4) further addresses how attributes of feedback regulation impact bistability. The analysis, shown in Figs. S2D and S3D, demonstrates that bistability does not hinge on high levels of cooperativity in RtcR-activation.

Comment 2.6: Nevertheless, the manuscript goes on to identify two of the three proteins of interest as potential targets to potentiate antibiotic effects.

Specific comment:

If it is known that the Rtc system is important for stress management or recovery from a stressed state and that it only comprises three proteins, targeting the entire Rtc system may be more effective than targeting just two members (RtcA and R). Since not all aspects of the Rtc system are fully understood, and for example that the Rtc system is known to affect mobility (as noted in a paper by one of the authors), it raises the question of why targeting the whole system might be considered inferior to targeting only RtcA and R. 3 out of 3 versus 2 out of 3

We consider targeting single components over the whole system, as currently no inhibitors for either of the Rtc proteins are known. Our analysis aims to identify, which of the three proteins may serve as the most effective target to suppress resistance, intended to inform the search for inhibitors.

Comment 2.7: Evidence,

So, the existence of bistability is almost guaranteed, which raises concerns about the novelty of this finding. Nevertheless, it is important to determine if some of the observed bistabilities are specifically due to the Rtc system. Understanding the role of the Rtc system in general bistabilities would be particularly interesting and valuable. However, this paper only hypothesizes bistability if the system is modeled in the specified way and with parameters from the literature. The authors state, "Our results suggest that the response may be underpinned by a form of hetero-resistance, and they suggest a granular characterization, at the level of single cells, will be necessary to test the hypothesis." The only evidence presented in this paper is the expression levels of the Rtc system (Supplementary Figure 1), which are very low.

The reviewer correctly points out that the experimental evidence presented in our previous submission was weak.

Thanks to the reviewer's comment, we now collected fresh data that provide evidence for our central model prediction regarding heterogeneity in *rtc*-expression and consequences to translational capacity as outlined below. The new data have substantially expanded the scope of our study and honed the implications of our findings.

The new analysis, summarised below, has led to substantial improvements in the revised manuscript, including a whole new Results section with a new main and supplementary Figure (Figs. 4 & S8), a new title, additional Methods sections, as well as additional text in the Introduction, Discussion and Abstract. As a result of the new analysis, we also included two additional authors who conducted the experimental work.

We opted to monitor mRNA expression of *rtcB* using single-molecule RNA FISH (Fig. 4). Consistent with previous data (Engl, Schaefer, Kotta-Loizou & Buck, 2016, Nucleic Acids Res., 44), induction through exposure to tetracycline increased mean expression (μ in Fig. 4C) by approximately 2.5-fold. Crucially, the distributions of mRNA copy numbers per cell display a large proportion of non-expressing cells in both induced and uninduced conditions, implying that average induction is driven by only a small proportion of expressing cells.

Our data indicate that average induction is a result of (a) higher copy numbers in expressing cells (z-test, p -value $< 10^{-3}$) and (b) a larger proportion of cells expressing Rtc. To show the latter, we employed a stochastic model of σ^{54} -controlled expression (Engl et al. 2020 Nat Commun 11, 2422) and accordingly fitted zero-inflated negative binomial distributions, which estimated 22%

vs 13% of activated cells in inducing vs uninduced conditions. Overall, the expression data validate that induction at the population-level is modulated via heterogeneity across cells, both, in terms of higher copy numbers and higher proportions of expressing or 'on' cells.

Moreover, to evidence the impact Rtc expression exerts on translational capacity, we quantified *in vivo* translation elongation speed using a LacZ α -complementation system. Figure 4 shows that lack of *rtcB* expression slows translation speed in both, induced and uninduced conditions.

As requested by reviewer #2, these new data provide evidence to validate the central hypothesis generated by our model that Rtc-mediated responses to ribosome-targeting antibiotics display population heterogeneity and impact translational capacity.

To our knowledge, our data represent the first quantification of Rtc expression at the level of single bacterial cells, which was entirely inspired by a model-derived hypothesis. We hope that our approach of model-guided discovery can stimulate new avenues for studies of RNA repair.

Comment 2.8: Specific comment:

5) I see how expression levels lower than 1 copy per cell indicate that some cells have 0 copies, but this does not necessarily indicate bistability. A gene that is very lowly expressed and does not show bistability could produce similar results.

We appreciate this comment and share the reviewer's concern that our previous qPCR data only provided weak evidence for heterogeneity in *rtc* expression. With our new single-cell data providing stronger evidence, we now removed the supplementary figure depicting the previous qPCR data mentioned in this comment and only kept values that were used to tune predicted expression levels (see Table S4).

We note that aside from bistability other factors may also influence levels of observed heterogeneity, some of which we mention in our response to Comment 2.10 below. Given the low levels of *rtc* expression, stochasticity is also likely to contribute to observed heterogeneity – we included a comment on this in the Discussion. A comprehensive exploration of these factors is certainly desirable but exceeds the scope of the current study.

Our focus with the new data was on evidencing heterogeneity in expression and its downstream effects, same as other studies before us did, e.g. the study on innate growth bistability cited by the reviewer (Barrett Deris et al. 2013 Science 342.6162). We acknowledge that the smRNA-FISH data is also inconclusive on bistability, however, it provides strong evidence of the key conclusion we drew from our model analysis, namely that cell-to-cell heterogeneity can arise from the mechanisms underlying Rtc-mediated RNA repair. We believe that the discovery of heterogeneity in an antibiotic response, in and by itself, is more impactful than whether the system exhibits hysteresis within a measurable range of antibiotic conditions.

Comment 2.9: The model is not used to emulate any experimental data (levels of Rtc members, etc).

Specific comment:

The model is not validated any experimental data? How is this model validated?

Above we explained how our new data provides strong validation for the conclusions drawn from our model analysis regarding heterogeneity arising from the regulation of the Rtc system. Additionally, qPCR data was used to tune *rtc* expression levels of the model, so we have

informed parameter choices for the unknown parameters controlling gene expression (ω_{ba} and ω_r). The revised manuscript now includes an explanation of how the qPCR data was used (see §S2.1).

Comment 2.10: Since there not much experimental evidence, I have looked elsewhere. I have been troubled with the conflict with experiments in other papers and the presumed role of Rtc system.

Specific comment:

7) In my understanding of this manuscript, the wild-type bacteria with an Rtc system should show bistability. However, according to experiments in “Deris, J. Barrett, et al. ‘The innate growth bistability and fitness landscapes of antibiotic-resistant bacteria.’ Science 342.6162 (2013): 1237435,” the wild-type bacteria do not show bistability when treated with chloramphenicol (which is also indicated to induce the Rtc system, as reported in Engl, Christoph, et al. ‘Cellular and molecular phenotypes depending upon the RNA repair system RtcAB of Escherichia coli.’ Nucleic Acids Research 44.20 (2016): 9933-9941). How can we reconcile these two seemingly conflicting results?

We thank the reviewer for this valuable comment. We cannot say for certain why the two papers obtained seemingly conflicting observations, however, several differences in the experiments may have led to this result.

First, Barret Deris et al. measured heterogeneity in terms of growing vs non-growing cells, whereas susceptibility was measured as alteration of growth rates by Engl et al. (note that the conditions we consider are sub-inhibitory, and tetracycline as well as chloramphenicol are bacteriostatic drugs). It is therefore possible that differences in growth rate were too subtle to be detected when the focus was on growth vs growth-arrest.

Second, the above argument may be enhanced by the fact that Rtc-expressing cells only represent a small minority of cells (see Fig. 4), and thus may not have been detectable by the approach employed in Barrett Deris et al.

Third, we highlight that the experimental conditions between the two studies differed in terms of growth medium – Barrett Deris et al. used minimal media, and Engl et al. LB – as well as antibiotic concentrations tested.

Our revised manuscript includes a comment on this seeming contradiction in the Discussion.

Comment 2.11: Relevance,

According to one of the cited papers, “Manwar, Muhammad Ramzan, et al. ‘The bacterial RNA ligase RtcB accelerates the repair process of fragmented rRNA upon releasing the antibiotic stress.’ Science China Life Sciences 63 (2020): 251-258,” the authors state: “Therefore, rtcB is widely distributed but not highly prevalent across bacterial lineages, and such features suggest possible non-essential physiological functions which are perhaps advantageous under specific environmental conditions”. The prevalence that they give are very low.

Specific comment:

If the RtcB is not very prevalent, how can targeting it would be clinically relevant

We addressed this concern in our response to Comment 1.1, where we limited our conclusions to *E. coli* and its closely related species that carry *rtcBA* arranged in an operon and under the regulation by RtcR. We note, however, that the concerned species include nosocomial pathogens of critical clinical relevance due to their elevated levels of antibiotic resistance.

Comment 2.12: Overall,

In my understanding, after reading other papers by the authors (who are experts on Rtc), I am persuaded that bistability may exist for some bacteria in response to certain drugs, and it may be important to understand the role of the Rtc system in this context. However, the evidence remains hypothetical. Although the manuscript is eloquently handled and represents a good modeling effort, it lacks substantial validation and evidence, remaining a theoretical exploration that highlights the potential importance of the Rtc system.

We thank the reviewer for this largely positive assessment of our manuscript. We hope that the experimental validation we now present satisfactorily addresses the reviewer's remaining concern regarding contributions of our work beyond theoretical exploration.

Responses to comments by Reviewer #3:

Comment 3.1: This is a very interesting paper, developing a mathematical model of the Rtc RNA repair system, which the authors (convincingly) argue could be used in the fight against antibiotic resistance. Their findings suggest several predictions to be tested experimentally, as well as proposed treatments against bacterial infections.

The paper is well-written, the maths are clear and the figures are pretty. However, I have several questions I'd like the authors to address:

We thank the reviewer for this favourable assessment of our manuscript.

Comment 3.2: What happens if RtcB can work directly on damaged ribosomes, as you mention in line 161, page 2? It seems that it wouldn't change much qualitatively, right?

This is correct – we addressed this concern in our response to Comment 1.3, where we added a new model analysis of this specific scenario. The analysis demonstrated a dynamic response that is largely indistinguishable from the main scenario we consider, where RtcA is needed for concerted RNA repair.

Comment 3.3: In eqs. 1-3, shouldn't growth rate be a function of healthy ribosomes, according to Scott and Hwa's growth laws, or the corresponding author's own model published in PNAS in 2015? As healthy ribosomes go down, so does growth rate, and this introduces a nice feedback that could lead to a nice expression for growth rate as a function of the parameters (similarly to what Greulich et al did using these growth laws: <https://www.embopress.org/doi/pdf/10.15252/msb.20145949>).

In fact, if you take v_{in} , v_{rep} and v_{tag} as linear, and write $r_h = \lambda/K_t$ (Scott's first growth law) then you can find expressions for all three ribosome species as a function of λ and the repair parameters (which are a function of p_A and p_B and thus of growth rate as well, in equilibrium), and sum all three ribosome equations to obtain one polynomial equation for growth rate as a function of the parameters. You don't need to do this last part, but I'd like to see how the model changes if you include the feedback between ribosomes and growth rate. Or at least mention this as an extension of the model in the Discussion.

We thank the reviewer for this valuable comment, which was similarly raised by reviewer #2 (see Comments 2.2 & 2.3). In response, we included a new section in the SI (S1.4) that specifically

addresses the assumption of dynamically linked growth/dilution rates, as well as ribosomal influx, to the availability of ribosomes – similar to what Greulich et al did.

Notably, in this analysis we assumed a negative correlation between growth rate and ribosome levels, as the scenario we consider, where growth is inhibited by translation-targeting antibiotics, falls under Scott's second growth law ($r = r_{max} - \lambda/\kappa_n$, Scott et al 2010 Science 330-6007). Scott and others (including our ourselves) ascribed the negative correlation to the levels of ribosomal inhibition, to which cells respond with increased levels of ribosome synthesis.

Since our model of RNA repair does not explicitly consider the mechanistic actions of ribosome inhibition (other than damage), it appears uncertain how other rates, such as v_{rep} and v_{tag} , should be most suitably linked to ribosome levels. We therefore prefer to avoid rigid assumptions in the model presented in the main, however, the new supplemental analysis addresses the issue in parts. We included a statement in the Discussion addressing interactions with dynamic growth physiology.

Comment 3.4: The main text should explain that v_0 in Figure 2B is a linear function of p_{R^} . The authors explain this with words in the main text, and show the equations in the Supp, but adding the equation for v_0 in the main text after the explanation (in line 275) would clarify things.*

This has been fixed now.

Comment 3.5:

- *The references to Cleland 1975 (ref 4 in Supp) has errors. Also ref 16 has too many caps!*
- *Eq 18 in the Supp should end with r_h , not r_d*
- *At the beginning of page S3, the definitions of K_a and K_b are reversed.*

Thanks for spotting those mistakes – now fixed.

Comment 3.6: The organization of Supp Figures is confusing. I suggest getting rid of S5A, which is not very informative (protein levels are proportional to mRNA levels, and so there is no need to show this, I think). S5B (which is not mentioned in the main text) seems to be identical to 3B, and should be removed. Then the text naturally cites S1 first. Supp Figure S5C should become S2, and the actual S2 would then become S3. Supp Fig S5D should become a separate figure, S4. Current figures S3 and S4 would then become S5 and S6.

Thanks for spotting these potential sources of confusion. We now rearranged supplementary figures to match their appearances in the main text.

Re S5A (now S2C): since this might not be obvious to every reader and since this is a supplementary figure, we prefer showing steady state mRNA levels for completeness.

Fig. S5B (now S3A) depicts results of sensitivity analysis with respect to different values of ω_r instead of ω_{ab} - we modified the caption to clarify.

*Comment 3.7: In L368, "This finding is backed-up by gene expression data (Figure S1), where average molecule numbers of the *rtc* mRNAs are well below one in WT and inducing conditions, suggesting that in a population of cells, not all cells are expressing *rtc*. In inducing conditions, upon exposure to ribosome-targeting antibiotics, this could suggest a heterogeneous response."*

Where is this data from? Is this from your lab? I don't see any description of the Methods used.

Apologies for this oversight – we have now included the qPCR data acquisition in the SI (§S.1).

Comment 3.8: In Fig 3C, what are the values of lambda and omega_ba used in the first plot, the values of ATP and omega_ba in the second plot, and the values of ATP and lambda used in the third plot? Also, this figure looks too crowded, I'd suggest adding more padding between subplots. Alternatively, make it a double-column figure.

Apologies for the ambiguity. Unless otherwise stated, the parameter values are those given in Table S1. We added a comment to the caption of Figure 3 to clarify this.

Comment 3.9: In Fig S5C, the grey lines should be dashed in the unstable branch.

Thanks for spotting this – now fixed.

Comment 3.10: I'm not sure I understand Fig S2. You claim that 99% of parameter combinations yield bistable responses, but for all parameters, the area of the violin plots yielding monostable responses is the same as the area yielding bistable responses? Is this a mistake?

We now amended the violin plots to display absolute frequencies rather than relative frequencies to illustrate the high occurrence of parameters yielding bistable responses.

Comment 3.11: In Fig 4A, I find it strange that the values leading to the OFF state are so few. I would have expected all initial conditions below the unstable branch should lead to the lower stable point. Why doesn't this happen? Can you do a phase-plane analysis to show this?

This would be true if we changed initial values of all variables simultaneously, however, we only change one variable at a time. Switching steady states therefore requires stronger perturbation, and hence fewer values achieve this. We added a comment to clarify this at the beginning of the perturbation analysis.

Comment 3.12: In L547: "Finally, we assumed that the rate of influx of healthy ribosomes is proportional to the rate of translational activity exerted by ribosomes themselves." What is the rationale behind this?

We acknowledge that this statement was unclear and without sufficient context. Our rationale was that synthesis of new ribosomes is energy-dependent and thus will slow down at lower ATP level, most notably, through energy-dependence of translation of ribosomal proteins. Given that influx rate was generally saturated for ATP concentrations of interest, and moreover, with our new supplemental analysis of dynamic influx covering dependency on ribosomal activity (see §S1.4), we removed this dependence and now consider a constant influx rate k_{in} (see Eq. 1).

We thank the anonymous reviewers for continuing to provide their time and expertise. Their constructive feedback has greatly improved the clarity of our manuscript and led to additional results that substantially enhance our conclusions.

Importantly the revised manuscript now reports new data that provide a strong initial validation of our predictions regarding differential effects of Rtc inhibition. The data is presented in an additional main figure (Fig. 6) and a substantive extension of the final results section. This additional work is now clearly reflected in the revised abstract, introduction and discussion and led to the inclusion of two new authors.

All changes are highlighted in the revised submission, and for convenience, we also included screenshots of the relevant sections in our point-by-point response below.

We hope the reviewers will be satisfied with how we thoroughly addressed their concerns, and we look forward to receiving their feedback.

Reviewer #1:

The authors refer in the Introduction to “Rtc” as a “system” comprising both RtcB (ligase) and RtcA (cyclase). They then say that “Rtc” functions in metazoan/archaeal tRNA splicing. But this is not strictly the case, insofar as the RtcB ligase is clearly a tRNA splicing enzyme in those taxa but (as far as this reviewer is aware) there is no evidence that RtcA plays any role in tRNA splicing.

We have rephrased the paragraph for clarity and to match with revisions we previously made regarding *rtcBA* acting as a conserved “system” with core genes *rtcR*, *rtcBA* only in *E. coli* and closely related species:

¹⁹ Here we investigate two RNA repair proteins, the RNA
²⁰ cyclase RtcA and the RNA ligase RtcB, which are conserved
²¹ across all domains of life [7]. In metazoans and archaea,
^{le} RtcB plays a well-defined role in tRNA ligation after intron
^{rtb} excision and in the unfolded protein response [8, 9]. In
ⁱⁿ bacteria, however, the exact roles of Rtc are less known.

I had previously objected to central assumption of the model, as depicted in Fig. 1B, that the RNA damage that incises the phosphodiester backbone leaves a 3'-PO₄,2'-OH end that must be cyclized by RtcA to form a 2',3'-cyclic phosphate end that serves as the substrate for end joining by the RtcB ligase. This is still not a valid assumption. RtcB is just as active in sealing a 3-PO₄ end as a cyclic PO₄ ends. In their rebuttal, the authors cite published findings that the rate of single-turnover 3-PO₄ cyclization by RtcA is much faster than the rate of ligation by RtcB as a validation of their assumption that any 3'-PO₄ end will be cyclized prior to encounter with RtcB. But this is off point. Single-turnover kinetics are performed in enzyme excess. In the cellular milieu, the partition of 3'-PO₄ ends between cyclization and ligation will depend on the concentrations of RtcA and RtcB (not known?) and the relative affinities of RtcA and RtcB for their respective 3' end substrates in the context of damaged tRNAs and rRNAs of concern in this paper (also unknown?).

Therefore, I consider the statement on lines 177-180 (“When the two are co-expressed, as in the antibiotic response we consider, RtcA thus likely removes reactive 3'-phosphate ends before RtcB can, and so in the model we only consider the action of RtcB on 2',3'-cyclic phosphate

termini.”) as unjustified.

Concerted repair by RtcAB is currently accepted as a credible repair path, see e.g. a recent influential paper on the subject (Uppalapati et al. *Science* 2024), which the reviewer also highlighted in their earlier review. The comment moreover overlooks our analysis of RtcA-independent repair by RtcB, so long as an RtcR-activating ligand is present (Supplementary Information §S1.3.3 and Fig. S1):

S1.3.3 Direct repair via RtcB only

It is possible that some RNA damage may not produce 3' RNA ends but infact produce 2',3'-cP ends [8]. In this scenario RtcB is able to directly repair 2',3'-cP ends, and RtcA is not required. We model this scenario by removing RtcA mRNA and protein species and removing damaged RNAs. Therefore, tagged RNAs are directly produced by damage at rate k_{dam} , and we no longer have a 'tagging' step in the repair. The scenario yields a modified equation for tagged RNAs

$$\dot{r}_t = r_h \cdot k_{\text{dam}} - v_{\text{rep}} - \lambda \cdot r_t. \quad (\text{S27})$$

Figure S1 shows a comparison of the dynamic responses when considering damage that directly yields 2',3'-cyclic phosphate ends, which can be repaired by RtcB without action by RtcA, and when considering damage that yields 3'-phosphate RNA ends, which require the concerted action of RtcA and RtcB. Steady-state response between the two model versions are largely equivalent, i.e. bistability in *rtc* expression and translational capacity with largely the same steady-state values except for minor deviations in a small range of damage rates (Figure S1 top). There are also minor differences between the two scenarios in their transient behaviour, as steady states are approached more rapidly when damage directly yields the ligand that induces *rtc* expression (Figure S1 bottom).

Figure S1: **Direct repair via RtcB only.** Comparison between scenarios where damage yields 2',3'-cP RNA ends (direct repair by RtcB only) and 3'-P termini (concerted repair by RtcBA as considered in the main models) for ribosome (rRNA) repair (A) and tRNA repair (B). RtcB (purple) is displayed on the left *y*-axis and healthy RNA (orange) is displayed on the right *y*-axis. Darker shades of purple or orange represent predictions for repair of 3' ends and lighter colours represent predictions for direct repair of 2',3'-cP ends via RtcB only. (Top row) Bifurcation diagrams display minor differences in steady-state values between 3' repair and the 2',3'-cP repair for rRNA and no difference for tRNA. (Bottom panels) Time-resolved trajectories at exemplar damage rates demonstrate that steady states are generally reached more rapidly in the case of 2',3'-cP repair. Differences are minimal in the tRNA model at all damage rates.

These results were included in our previous revision and indicate that *rtcBA* regulation promotes heterogeneity in Rtc-mediated repair, and so our conclusions hold irrespective of whether or not RtcA is involved in the repair. Our consideration of various repair routes lends greater generality to the conclusions of our analysis. The request to narrow our assumptions therefore appears unnecessary, especially, given that there is currently no established consensus in the literature.

Regarding physiological concentrations, we recognize that intracellular concentrations can differ from enzymatic assay conditions – this is a general challenge for *in vivo* characterization of protein function – but it is expected that RtcA and RtcB concentrations are of comparable magnitude given their co-expression. Further, it is true that affinities are likely affected by the cellular context of damage amongst physiological substrates when compared to single-turnover assays with clearly defined substrates. But given the clear evidence that RtcA acts markedly faster on readily accessible RNA ends that represent damage, we cannot assume that occlusion by one or more unknown cellular contexts would solely affect RNA substrate binding by RtcA, and not by RtcB.

The heart of the matter is whether the cyclase activity of RtcA is necessary for RNA repair by RtcB, as the authors predict and assume in Fig. 1B. The mere fact that the rtcB and rtcA genes are coexpressed and co-induced does not establish this point. The authors state in rebuttal that “ΔrtcA and ΔrtcB strains display impaired growth compared to wildtype E. coli, and complementation with RtcA or RtcB, respectively, recovers wildtype growth (Engl, Schaefer, Kotta-Loizou & Buck, 2016, Nucleic Acids Res., 44), implying that both are required for full growth rescue in the conditions we consider.” However, the prior NAR paper does not establish the point regarding RtcA. For example, the NAR paper did not establish whether the phenotypes attributed to ΔrtcB and ΔrtcA reflect lack of the proteins versus lack of their RNA repair catalytic activities. RtcA and RtcB both act via a mechanism of covalent nucleotidyl transfer to an active site histidine nucleophile. The active sites have been defined, so it would be straightforward to compare complementation of ΔrtcB and ΔrtcA phenotypes by wild-type and His-to-Ala mutant alleles of rtcB and rtcA. The NAR paper demonstrated fragmentation of rRNA in ΔrtcB cells (presumably owing to a defect in rRNA repair) that was rectified by expression of RtcB. I will grant that it is overwhelmingly likely that an active site mutant of RtcB will not complement this phenotype. Yet no such analysis was presented for ΔrtcA cells so we are in the dark as to whether lack of RtcA elicits rRNA fragmentation (an rRNA repair defect) and whether this is because of a lack of cyclase activity. (In this vein, it is worth noting that expression of E. coli RtcB in yeast in lieu of yeast Trl1 complements tRNA splicing and Hac1 splicing in the UPR, in the absence of any RtcA.) Thus, I do not agree with the authors’ assertion that “our assumption that RtcA has a role in the repair response seems well justified.”

Previous work (Engl et al, NAR 2016) evidenced the need for RtcA in the phenotype of interest: ΔrtcA strains could not recover growth to the same extent as WT strains could when challenged with an antibiotic that targets the ribosome, whereas complementation with wildtype RtcA restored growth recovery. Complementation with a catalytic His-to-Ala mutation was not tested in the antibiotic scenario, however, in all the other rtc-inducing conditions it was tested (including exposure to the ribotoxin colicin D and starvation, see Figs. S1c & S2 in Engl et al), complementation of ΔrtcA with the His-to-Ala mutation produced growth phenotypes that were equivalent to the zero deletion whereas complementation with wildtype rtcA restored wildtype phenotypes. The evidence therefore indicates that the likely basis of the ΔrtcA phenotype is the lack of rtcA-encoded cyclase activity on the damaged RNA end.

Our previous revision further showed that our main conclusion regarding bistability/heterogeneity is agnostic to whether tRNA or rRNA is the repair substrate – our analyses cover both scenarios, as well as each of them with a concerted RtcAB (Figures 3 & S2, see below screenshots) and an RtcA-independent repair mechanism (Figure S1, see above).

Figure 3: Steady-state responses of the rRNA-repair model.

Figure S2: Steady-state responses of the tRNA model. (A) Stability analysis: Bifurcation dia-

With regards to the relevant physiological substrates, these are still unknown and, as highlighted by Uppalapati et al. Science (2024), the identity of substrates may be stress specific. While of clear scientific interest, it is beyond the scope of our manuscript to establish the precise substrates and their differential repair routes. Instead, our paper focuses on the effects of various known aspects of Rtc-mediated repair, including competing hypotheses around them, all of which support our main conclusion – that *rtc* regulation promotes heterogeneity in an antibiotic response, which was previously unknown and which we validated experimentally.

The authors acknowledge (lines 167-170) that RtcA can also convert 2'-PO4 ends into a cyclic-PO4 that can be repaired by RtcB, but they dismiss this reaction from consideration in their model because it is much slower than 3'-PO cyclization. This seems arbitrary and perhaps unwise. As noted above, there is no a priori reason for RtcB to “care” whether it is ligating a 3'-PO4 or cyclic-PO4 end. But if RNA damage produces a 2'-PO4 end, either directly or when the E. coli CPDase ThpR converts a cyclic-PO4 end into a 2'-PO4 end (PMC4201822), there is no way for this 2'-PO4 end to be sealed by RtcB unless RtcA cyclizes it.

This scenario is equivalent to the main scenario we consider, namely that RtcA and RtcB engage in a concerted repair response, but at a much-reduced repair rate. We now included this scenario in Fig S5, which shows that bistability is less robust and displayed only in more extreme parameter regimes. Given that heterogeneity was indeed observed experimentally, this may indicate that 3'-repair is the more likely repair substrate in the conditions we consider.

180 We primarily model RtcA action on 3'-phosphate ends, but
 181 also consider the action on 2'-phosphate ends by using the
 182 lower catalytic rate of cyclisation in the model. 236

459 To account for the scenario where RtcA acts on 2'-phosphate
 460 ends, albeit at a slower rate (0.1 min^{-1}) to 3'-phosphate termi-
 461 ni, we decreased the k_{tag} parameter (SI Figure S5). We
 462 observe that at the slower rate of tagging of 2'-phosphate
 463 termini for both rRNA and tRNA repair the model displays
 464 bistability at extreme parameter values and is less robust
 465 than bistability observed in 3'-phosphate repair.

Figure S5: The cyclisation of 2'-phosphate RNA termini by RtcA gives a bistable response in extreme parameter regimes. For both the rRNA and tRNA models, the mechanism of repair of a 2'-phosphate termini via RtcA ($k_{\text{tag}}: 0.1 \text{ min}^{-1}$), leads to a monostable response in the parameter regime presented in Table S1 (grey curves) and a bistable response in an adjusted parameter regime (purple curves). The adjusted parameters are as follows: $\lambda: 0.0038 \text{ min}^{-1}$, $\omega_{ba}: 8 \cdot 10^{-4} \text{ min}^{-1}$, $\omega_r: 2 \cdot 10^{-5} \mu\text{M} \cdot \text{min}^{-1}$. rRNA model specific parameters: $k_{\text{in}}^{\text{rRNA}}: 0.4 \mu\text{M} \cdot \text{min}^{-1}$, tRNA model specific parameters: $k_{\text{in}}^{\text{tRNA}}: 4 \mu\text{M} \cdot \text{min}^{-1}$, $R: 80 \mu\text{M}$.

This highlights a central knowledge gap in that we do not know what types of RNA ends are generated during the damage in the antibiotic response that the authors invoke. In many well-studied instances (ribotoxin endonucleases that incise RNA via transesterification), the immediate product is a cyclic-PO4.

We agree, this is why we included an analysis of the latter scenario in §S1.3.3 in our previous revision.

S1.3.3 Direct repair via RtcB only

It is possible that some RNA damage may not produce 3' RNA ends but in fact produce 2',3'-cP ends [8]. In this scenario RtcB is able to directly repair 2',3'-cP ends, and RtcA is not required. We model this scenario by removing RtcA mRNA and protein species and removing damaged RNAs. Therefore, tagged RNAs are directly produced by damage at rate k_{dam} , and we no longer have a 'tagging' step in the repair. The scenario yields a modified equation for tagged RNAs

$$\dot{r}_t = r_h \cdot k_{\text{dam}} - v_{\text{rep}} - \lambda \cdot r_t. \quad (\text{S27})$$

Figure S1 shows a comparison of the dynamic responses when considering damage that directly yields 2',3'-cyclic phosphate ends, which can be repaired by RtcB without action by RtcA, and when considering damage that yields 3'-phosphate RNA ends, which require the concerted action of RtcA and RtcB. Steady-state response between the two model versions are largely equivalent, i.e. bistability in r_{tc} expression and translational capacity with largely the same steady-state values except for minor deviations in a small range of damage rates (Figure S1 top). There are also minor differences between the two scenarios in their transient behaviour, as steady states are approached more rapidly when damage directly yields the ligand that induces r_{tc} expression (Figure S1 bottom).

The authors have tempered their assumptions that rRNA repair is the prime function of RtcAB and now extend the activity to spectrum to broken tRNAs (for which there is more persuasive evidence). Yet I still have difficulty accepting their classification of ribosomes (or tRNAs) as healthy (unbroken), damaged (3'-PO4, 2'-OH breaks), or tagged (2',3'-cyclic phosphate ends formed after the action of RtcA), whereby the tagged states can be repaired by RtcB. I do not see the justification for invoking the "tagged ribosome" (or tagged RNA) state as the consequence of RtcA action on "damaged" ribosomes (or tRNAs) with 3'-PO4, 2'-OH breaks, as opposed to such ends being "damaged" themselves.

We recognise our use of potentially confusing language. By 'tagged' we denote an RNA end that is still damaged but has been pre-processed for final repair. In the current revision we clarified our definition of 'tagged':

SI). To this end, we model three distinct species of ribosomes: healthy ribosomes that can obtain breakage within their rRNA and become damaged ribosomes; upon interaction with RtcA, damaged ribosomes convert to what we call ‘tagged’ ribosomes, **which are still considered to be damaged but have been pre-processed for final repair. These tagged ribosomes** contain cyclized rRNA ends and serve as a substrate to RtcB for full repair to recover healthy ribosomes.

*The authors include new experiments that image single cells for *rtcB* expression by FISH before/after exposure to tetracycline and indicate that there is cell-to-cell heterogeneity whereby only some cells are induced. They provide evidence that translation speed is reduced in $\Delta rtcB$ cells in the absence of antibiotic. What really needs to be done here is to perform the same translation speed experiment for $\Delta rtcA$ cells (while also doing the rRNA damage assay on $\Delta rtcA$ cells a la the NAR paper; vide supra). In other words, test the assumption that RtcA (and its cyclase activity) is important as posited.*

Our revision covers computational analyses of every mechanistic scenario that was suggested by the reviewer, all of which yielded the same insights. Our conclusions do not hinge on the involvement of RtcA; our analyses indicated that, irrespective of the precise repair path (concerted vs RtcA-independent), the current body of knowledge surrounding Rtc-regulation points to the conclusion that it promotes heterogeneity, which we validated experimentally in the previous revision.

We acknowledge the importance of establishing greater mechanistic clarity. However, since the issue was not raised previously, in agreement with the Editor, we decided to focus additional experiments on addressing comments that more directly target the validity of our conclusions.

*A concern here is that it is not obvious how to account for the global slowing in translation in $\Delta rtcB$ cells versus wild-type given that the vast majority (~90%) of wild-type cells have no *rtcB* mRNA (detectable by FISH) to begin with. Could it be the case that the FISH assay underestimates the mRNA level in individual cells scored as having 0 transcripts.*

Elongation speed was measured in bulk and therefore yields a population average speed. We are unsure whether the reviewer expected a more drastic or less pronounced average slowdown in $\Delta rtcB$ strains. In the single-cell quantification of *rtcB* mRNA expression, we calibrated intensities with negative $\Delta rtcB$ controls following standard protocol. The obtained measurements are also consistent with the qPCR data (Table S4) that indicate average copy numbers well below zero, implying that far from all cells express *rtc*.

I had raised a question about the allosteric model for activation of the RtcR hexamer. The authors state (lines 281-284) that this requires “the cooperative binding of six ligands [RNA with 2',3'-cyclic termini] for full activation.” Why can't allostery be achieved by binding of one or two ligands? The statement in the rebuttal (“the reviewer is correct that we currently do not know how many ligands are required for full activation, and thus we cannot categorically state that six ligands are used by one RtcR hexamer”) seems to be at odds with the categorical statement/assumption in the Results.

In our previous revision, we performed a detailed sensitivity analysis for the allostery coefficient (Figs. S3D & 4D) and concluded that our results are not limited by the hexamer assumption (Results, lines 452-459). We recognize our use of too categorical language when first introducing the assumption and amended this in the current revision:

293 Consequently, in the model we assume that RtcR acts as
 294 a hexamer, as is typical for bEBPs [43], therefore requiring
 295 the cooperative binding of up to six ligands for full acti-
 296 vation [19, 44]. We model this cooperative binding with

Reviewer #2:

Summary

The final version of the paper argues that a model of the Rtc system supports robust bistability (heteroresistance). This conclusion is reinforced by new FISH experiments showing that only a small fraction of bacteria express Rtc; upon antibiotic exposure, however, this fraction grows. Moreover, $\Delta RtcB$ cells exhibit slower translation rates both with and without antibiotics, with a “slightly more pronounced effect in the presence of antibiotics,” as argued by the authors. The authors further suggest that inhibiting RtcB and RtcR might be more effective in blocking this effect than targeting RtcA.

Overall, there is clear progress from the initial submission to the revised manuscript. Credit to the authors that the new experiments and their results are important on their own. However, I think there are several major issues.

First of all, throughout the manuscript, the model is used for its qualitative insights: its demonstration of bistability supports the contention that bacterial cells can exhibit heterogeneous responses.

An alternative approach could have involved systematically measuring certain rate constants or other things, comparing them against predictions, and thereby validating the model more rigorously. While such a quantitative approach may not be strictly necessary, it underscores the qualitative (rather than quantitative) approach of the manuscript.

We thank the reviewer for their positive assessment regarding the progress and importance of our results. Regarding the qualitative nature of insights drawn from model predictions, however, we note a discrepancy between the previous and current assessment by the reviewer:

In our first submission, the reviewer highlighted the methodological soundness of our mathematical modelling work, and their main concern was experimental validation of the conclusions on heterogeneity drawn from the models, not validation of the models themselves. In response, in our previous revision, we indeed validated the conclusions experimentally, and we are glad that this was now positively appraised by the reviewer.

While we recognise the value of quantitative validation or parameterization with fresh data, for the current study, it is unclear what specific insights they can add to our conclusions and what level of validation will be deemed sufficient. In discussion with the editor, we jointly agreed that further validation should instead focus on predictions regarding Rtc-inhibition (see response to comment 5 by the reviewer below).

Since the model is not validated quantitatively, it should be validated qualitatively. These considerations raise two important questions:

1. *Do the authors fully stand by the claim of bistability as predicted by their model?*
2. *Do they assert that blocking RtcR and RtcB indeed leads to significant potentiation of antibiotic treatment? and Rtc A does not.*

I consider that authors ought to satisfactorily answer these questions upon the qualitative treatment of the model. I will elaborate while going over the points that are answered by the reviewers.

We respond to these questions when they are elaborated with more detail below.

1. Assumptions on Growth Rate and Ribosome Influx

Addressed satisfactorily.

2. Cooperativity and Positive Feedback in RtcR Activation

The claims are toned accordingly.

We are encouraged by this positive assessment.

3. Experimental Validation and Evidence for Heterogeneity

This ties into the first major question mentioned in the summary. Does the data strongly support bistability? Low expression levels can lead to most cells lacking mRNA copies and some having a few copies. It is not clear to me as it is. Can authors claim bistability from their data definitively? Or further experimentation is needed.

We partially addressed this issue in our previous rebuttal (Comments 2.8 & 2.10). To avoid any ambiguity on the nature of evidence provided, we now included these points in the new revision.

In summary, heterogeneity in an antibiotic response is a more important discovery than whether the system exhibits hysteresis within a measurable range of antibiotic conditions. As with other studies, including the study on innate growth bistability cited previously by the reviewer (Barrett Deris et al. 2013 Science 342.6162), our focus was thus on evidencing heterogeneity rather than bistability, which would have required evidencing hysteresis in addition to heterogeneity. We acknowledge that the smRNA-FISH data is inconclusive on bistability, however, it provides strong evidence for cell-to-cell heterogeneity, which was the key conclusion we drew from our model analysis.

Aside from bistability, other factors may also influence levels of observed heterogeneity. For example, given the low levels of *rtc* expression, stochasticity is also likely to contribute – our previous revision included a comment on this in the Discussion, see below screenshot. A comprehensive exploration of these factors is certainly desirable but exceeds the scope of the current study.

gies to alleviate it. Possible limitations include the static nature of some of the physiological model parameters that are known to vary over time, which we addressed partially. Future work may build on our approach to investigate the role of a dynamic and adaptive cell physiology in shaping the adaptive resistance conferred by Rtc. Further, **our data on single-cell *rtcB* expression provided evidence of cell-to-cell heterogeneity, but not whether the cause of heterogeneity is indeed bistability, which would require observing hysteresis.**

We note that other factors also impact heterogeneity, for example, given the low levels of *rtc* expression we observed, stochasticity is also likely to contribute, raising interesting questions for future investigation. RNA repair is an area of

4. Clinical Relevance and Targeting the Rtc System

As mentioned in the manuscript, there may be many complementary mechanisms and even Rtc proteins that can replace the function of the studied Rtc system, but it is argued that this system

is the dominant one. This is a subjective assessment. It is not clear that blocking RtcB or R is enough to block tolerance of the cell under antibiotic exposure. This is indirectly shown by the fact that the $\Delta RtcB$ cell line has a slower translation rate. How can we be sure that it is enough? Can this be contrasted with anything that definitively causes tolerance, and its inhibition is clinically relevant?

Our assessment that RtcBA-mediated repair presents a dominant route to the observed resistance phenotype is based on evidence that WT strains can recover growth amid translation-targeting antibiotics, whereas $\Delta rtcA$ and $\Delta rtcB$ strains cannot (Engl et al, *NAR* 2016). The extent and clinical relevance of this form of resistance has yet to be established, however, many ribosome-targeting antibiotics are considered first-line antibiotics, potentiation of which is of clinical interest and importance. To add further support to the clinical relevance of Rtc-inhibitors, a study in *Salmonella enterica* recently highlighted a role of *rtcBA* in protecting the pathogen from oxidative stress exerted by the host immune response (Uppalapati *et al*, *Science* 2024; Svenningsen, *Cell Host & Microbe* 2024). We added a brief discussion of these aspects of Rtc biology to the Discussion of the revised manuscript.

ep^s limit the efficacy of inhibition. In view of these caveats, our
 an^s results motivate a host of experiments to assess the feasibil-
 for ity of targeting components of the Rtc system, the clinical
 uld relevance of which was also recently highlighted by work
 ibi^s that identified a role of RtcB in protecting *S. enterica* from
 vel oxidative host-immune assault [13].

5. Blocking RtcB and R is better than RtcA

Now the authors have shown that blocking RtcB does indeed lead to a lower translation rate (we still don't know if that is comparable to other literature-relevant cases). Why not try $\Delta RtcR$ and $\Delta RtcA$ and see if the response to $\Delta RtcA$ is different compared to $\Delta RtcB$ and $\Delta RtcR$?

We thank the reviewer for this comment, which prompted us to pursue fresh experiments and gather data that substantially strengthen the conclusions of our work. In the absence of known Rtc inhibitors, as suggested by the reviewer, our experiments set out to characterise the differential effects of individual deletions of *rtc* genes. We collected an exciting set of novel data presented in a new main figure (Fig. 6) as well as a comprehensive extension of the final Results section.

We quantified RNA/protein ratios, an established estimate for ribosomal contents, across individual *E. coli* cells. To our knowledge, RNA/protein ratios have previously only been reported at bulk, and so our data presents the first quantification at single-cell resolution. We used a new method based on Raman microspectroscopy, which is currently under consideration for publication, and which quantifies relative contents of abundant biomolecules including RNA, protein and DNA. This allowed us to compare distributions of ribosomal contents across wildtype and individual *rtc* knockout strains. In addition to average shifts in ribosome contents, we could therefore also investigate the impact on the cell-to-cell heterogeneity, enabling us to gain a granular view of the differential effects of individual *rtc* deletion.

We observed that $\Delta rtcB$ and $\Delta rtcR$ cells displayed significantly lower RNA/protein ratios than wildtype cells. Deletion of *rtcA* also resulted in a significant reduction of RNA/protein ratios, however, much less pronounced than the reduction caused by deletion of *rtcB* and *rtcR*. Lower ratios were due to lower RNA levels rather than higher protein levels. With cellular RNA dominated by ribosomal RNA, this indicates that cells lacking *rtc* maintain significantly lower ribosome levels, presumably because the reduced repair capacity means that more ribosomal

RNA is degraded. Interestingly, $\Delta rtcA$ cells displayed a bimodal distribution of RNA/protein ratios, where the higher mode coincided with the mode of the wildtype distributions and the lower mode with that of the $\Delta rtcB$ and $\Delta rtcR$ strains, indicating heterogeneity in ribosome levels with only some cells displaying lower ribosome contents than wildtype.

Overall, our new data provide strong validation of our model predictions that RtcB and RtcR, rather than RtcA, present promising targets to inhibit the repair actions upon translational antibiotic assault. We therefore hope that the reviewer will see their concern addressed to full satisfaction.

Figure 6: **Deletion of individual *rtc* genes decreases ribosome levels in tetracycline-treated cells.** (A) Mean RNA/protein levels, plotted with 95% confidence intervals, are significantly lower than the WT for each knockout strain (WT: $1.03 \in [1.00, 1.06]$, $\Delta rtcA$: $0.97 \in [0.94, 1.00]$, $\Delta rtcB$: $0.91 \in [0.88, 0.93]$, $\Delta rtcR$: $0.84 \in [0.82, 0.85]$). $\Delta rtcB$ and $\Delta rtcR$ have a stronger effect on the reduction in ribosome concentration compared to $\Delta rtcA$. Significance levels were obtained by performing unpaired two-sided t-tests for each knockout strain compared to the WT ($\Delta rtcA$ t-statistic: 2.88, degrees of freedom (df): 150, $\Delta rtcB$ t-statistic: 5.98, df: 143, $\Delta rtcR$ t-statistic: 10.25, df: 116), where *: < 5%, **: < 1%, ***: < 0.1%, ****: < 0.01%, ns: not significant. (B) Distributions of RNA/protein ratios indicate that all knockout strains display significantly reduced ribosome concentrations compared to WT, with levels in $\Delta rtcB$ and $\Delta rtcR$ lower than those in $\Delta rtcA$. (C) Distributions of total RNA and (D) total protein demonstrate that deletions have no significant effect on total protein concentrations but reduce total RNA. Asterisks indicate significance levels in a one-sided Mann Whitney U test comparing knockout levels to WT. For RNA/protein ratios (p -values $\Delta rtcA$: 0.008, $\Delta rtcB$: $7.9 \cdot 10^{-9}$, $\Delta rtcR$: $3.7 \cdot 10^{-19}$) and total RNA (p -values $\Delta rtcA$: $8 \cdot 10^{-4}$, $\Delta rtcB$: $4.1 \cdot 10^{-6}$, $\Delta rtcR$: $2 \cdot 10^{-4}$), we tested if knockout levels were smaller than WT, and for total protein (p -values $\Delta rtcA$: 0.991, $\Delta rtcB$: 0.997, $\Delta rtcR$: 0.736) if levels were greater than WT. (E) Histograms for each strain are shown with a kernel density estimate overlaid (solid lines). The distribution of the $\Delta rtcA$ strain displays two distinct modes that coincide with the modes of higher ribosome levels in WT (orange dashed line) and those of lower levels, i.e. more susceptible, in $\Delta rtcB$ and $\Delta rtcR$ (blue dashed line).

We then went on to seek experimental validation of our computational predictions. Since no Rtc inhibitors are currently known, we characterised translational capacity when expression of individual *rtc* genes is altered in cells that are exposed to a ribosome-targeting antibiotic. Using Raman microspectroscopy [?], we measured ribosome concentrations, by means of RNA/protein ratios, in wild-type *E. coli* and individual *rtc* knockout strains ($\Delta rtcA$, $\Delta rtcB$ and $\Delta rtcR$) at single-cell resolution (see Methods).

In the presence of tetracycline, all *rtc* knockout strains exhibited lower mean RNA/protein ratios than wild-type cells (Figure 6A). The data further revealed that, compared to wild-type, distributions across single cells display significantly lower RNA/protein ratios in $\Delta rtcB$ and, even more so, in $\Delta rtcR$ cells (Figure 6B, see Methods). While $\Delta rtcA$ cells also showed significantly lower RNA/protein ratios than wild-type cells, differences were not as highly

8

624 significant as in the $\Delta rtcR$ and $\Delta rtcB$ strains. We further 625 observed that lower RNA/protein ratios were due to lower 626 total RNA levels rather than increased total protein levels, 627 where differences were non-significant (Figure 6C & D). 628 628 Distributions of $\Delta rtcR$ and $\Delta rtcB$ RNA/protein ratios 629 displayed distinctly different modes from the wild-type 630 strain (Figure 6E). In contrast, $\Delta rtcA$ cells displayed a bi- 631 modal distribution, where modes coincide with those of the 632 wild-type and the other knockout strains (Figure 6E, red vs 633 blue dashed lines). Consistent with our model predictions, 634 the latter suggests that lack of rtcA leads to a heterogeneous	response where some cells are able to maintain full transla- tional capacity, similar to the wild-type, but some cells display impaired translational capacity, similar to that of $\Delta rtcB$ and $\Delta rtcR$. Consistent with our model predictions, our data therefore suggest that cells lacking rtcR and rtcB maintain lower transla- tional capacity, presumably due to impaired repair action, and deletion of rtcA is not as potent in locking cells in a state of lower translational capacity. Equivalent experiments per- formed in the absence of tetracycline display no significant differences in distributions between wild-type and all three
--	--

9

646 rtc deletion strains, indicating that this is a stress-specific 647 effect of the Rtc system induced by the antibiotic (Figure 648 S11). 649 649 Our results motivate further characterisation of the Rtc 650 system, in particular of RtcB and RtcR, to identify suitable 651 inhibitors and test their efficacy in rendering bacterial cells 652 susceptible to ribosome-targeting antibiotics. Given the 653 wide use of these antibiotics and strong conservation of the	rtcBA system across several clinically relevant pathogens, our results may point to a vigorous route for potentiating treatments and reducing levels of resistance.
---	---

10

Minor comments: There were some typos, repeated words; please proofread.

Apologies for this oversight – now fixed.

Reviewer #3:

Thank you for your detailed response. The authors have (more than adequately!) answered my queries. I recommend publication as is.

We are encouraged that Reviewer #3 highlighted the comprehensiveness of our revision.